



# Evolution and chemical characteristics of organic aerosols during wintertime PM2.5 episodes in Shanghai, China: Insights gained from online measurements of organic molecular markers

Shuhui Zhu[1,2], Min Zhou[1], Liping Qiao[1], Dan Dan Huang[1], Qiongqiong Wang[3], Shan Wang[2], Yaqin Gao[1],
Shengao Jing[1], Qian Wang[1], Hongli Wang[1], Changhong Chen[1], Cheng Huang[1,*], Jian Zhen Yu[2,3,*]

[1]State Environmental Protection Key Laboratory of the Cause and Prevention of Urban Air Pollution Complex, Shanghai
Academy of Environmental Sciences, Shanghai, China
[2]Division of Environment and Sustainability, Hong Kong University of Science and Technology, Hong Kong, China
[3]Department of Chemistry, Hong Kong University of Science and Technology, Hong Kong, China

*Correspondence to:* Jian Zhen Yu (jian.yu@ust.hk) and Cheng Huang (huangc@saes.sh.cn)

**Abstract.** Organic aerosol (OA) is a significant part of urban fine particulate matter (PM2.5) and a lack of detailed knowledge of their sources has increasingly hindered the improvement of air quality in China in recent years as significant reductions have been achieved in inorganic ion constituents. In this study, a wide range of organic molecular markers in PM2.5 were monitored with a bihourly time resolution using a Thermal desorption Aerosol Gas chromatograph system (TAG) in urban Shanghai in winter 2019. The molecular marker data have provided a unique source tracking ability in characterizing the evolution of organic aerosols during nine wintertime episodic events. Episodes primarily influenced by local air masses were characterized with higher proportions and mass increments of secondary OA. Rapid elevation in both absolute mass concentration and relative proportion was observed for primary and secondary OA markers indicative of vehicle emissions (e.g., alkanes, hopanes, and 2,3-dihydroxy-4-oxopentanoic acid), as well as cooking activities (e.g., saturated and unsaturated fatty acids, and C9 acids). In comparison, episodes under significant influences of transported air mass were typically associated with a predominant PM2.5 contribution from secondary inorganic aerosols and enhanced OA contribution from biomass burning activities. The latter was evident from the tracer data (e.g., levoglucosan, aromatic polycarboxylic acids, and nitro-aromatic compounds). Secondary OA markers associated with later generation products of hydrocarbon oxidation process, such as $C_{3-5}$ dicarboxylic acids, were the most deficient during local episodes while notably enhanced during the episodes under influence of transported air masses, reflecting different extent and pathways of atmospheric aging processing. The ability of distinguishing the variations of OA evolution during different types of episodes demonstrates the value of online organic molecular measurements to episodic analysis. The results indicate that control of local urban sources such as vehicular and cooking emissions would lessen severity of local episodes while regional control of precursors for secondary inorganic aerosols and biomass burning activities would reduce PM2.5 episodes under synoptic conditions conducive for regional transport.



## 1. Introduction

Fine particulate matter ($PM_{2.5}$) pollution has been one of the most prominent environmental issues in recent decades due to rapid industrialization and urbanization worldwide. In China, high concentration of $PM_{2.5}$ has resulted in significant drop of visibility (Zhang et al., 2012) and adverse impacts in mortality (L. Chen et al., 2017; M. Liu et al., 2017). Annual $PM_{2.5}$ concentration in China has been decreasing gradually over the past years with the implementation of a series of emission control measures focusing on reducing pollutions from energy usage, industrial processes and road transportation (Ding et al., 2019), however, haze episodes accompanied with abrupt elevation of $PM_{2.5}$ concentration still occur frequently in the cold season (Fan et al., 2021; Guo et al., 2020; Mao et al., 2019; Sun et al., 2019). According to the Report on the State of the Ecology and Environment (2019), on daily average, nearly six out of the 337 prefectural-level cities in China were under heavy or very heavy pollution. Among these high-pollution days, $PM_{2.5}$ as the leading pollutant took up 78.8%, which was much higher than the numbers of the high-pollution days with $PM_{10}$ and $O_3$ as the leading pollutants (Report on the State of the Ecology and Environment in China, 2019).

The accumulation of $PM_{2.5}$ is a combined result of source emissions, atmospheric dynamics, chemical transformation and wet/dry deposition. Many studies have shown that either local emissions or regional transport coupled with secondary processes under certain meteorological conditions are major contributors to short-term haze episodes in China (Cai et al., 2017; Chen & Wang, 2015; Huang et al., 2014; Li et al., 2016; Liu et al., 2017; Ren et al., 2014; Q. Wang et al., 2015; Y. Wang et al., 2014a; Zhao et al., 2013). Liu et al. (2014) and Q. Wang et al. (2015) investigated several cases of severe haze pollution in north China and identified that local traffic emissions together with enhanced coal combustion activities were the main causes of winter haze episodes. Wang et al. (2013) and Tong et al. (2020) also observed continuous new particles growth and subsequent secondary aerosol formation in the presence of strong biomass burning plume, suggesting that primary emissions from biomass burning could induce secondary formation of $PM_{2.5}$ and aggravate air pollution.

The significant contribution of secondary chemical transformation to winter haze episodes is also documented in the literature (Huang et al., 2014; Tao et al., 2017). Recent studies based on chamber experiments and observation measurements have provided solid evidence that both photochemical oxidation and aqueous phase transformation of gaseous precursors followed by gas-to-particle phase partitioning are important secondary processes during $PM_{2.5}$ episodes. Specifically, field campaigns in the Yangtze River Delta (YRD) region (Huang et al., 2020) and Beijing-Tianjin-Hebei region (Xiao et al., 2021) observed that aqueous-phase processing was more significant than photochemical oxidation in promoting the formation of more aged secondary organic aerosols (SOA) with higher oxidation state while photochemical gas-phase oxidation imposed larger impacts on the concentration level of bulk organic aerosols (OA). These field observations were in accordance with the results from chamber experiments (Chen et al., 2021; Hinks et al., 2018; Lim et al., 2010).

In this work, a field campaign was conducted in an urban site in Shanghai to characterize the evolution of haze episodes during the winter of 2019. During this campaign, 98 organic compounds in $PM_{2.5}$ were continuously monitored using a Thermal desorption Aerosol Gas chromatograph system (TAG) with a bihourly time resolution, along with hourly measurements of $PM_{2.5}$ major components and trace elements. The continuous online measurements of primary and secondary OA tracers by TAG have enabled observing episodic variations of organic aerosols, providing molecular level insights into formation and evolution of OA during winter haze episodes in urban atmosphere. In studying evolution processes, previous research predominantly deploys Aerosol Mass Spectrometer, which relies on molecular fragments for differentiation of OA sources. In comparison, this study has a unique advantage in source tracking in using more source specific organic molecular markers. Our observations reveal notable diversity in OA transformations between haze episodes under influence of different air masses,



providing measurement-based evidence in prioritizing control strategies for future air quality improvement.

## 2. Methods

### 2.1 Sampling site and online measurements

The winter campaign was conducted at the site of Shanghai Academy of Environmental Sciences (SAES) site (31°10′N,121°25′E) from 25th November 2019 to 23rd January 2020. Detailed descriptions of this urban site can be obtained in several published papers (Y. Liu et al., 2021; Q. Wang et al., 2020; He et al., 2020; S. Zhu et al., 2021). A comprehensive set of online instruments (Table 1) were operated on the roof of an eight-floor building (~25 m above ground) at SAES, including the TAG system (TAG, Aerodyne Research Inc). Additionally, multiple high-time-resolution instruments for the measurements of organic fragments in $PM_1$, major components and trace elements in $PM_{2.5}$, as well as gaseous and particulate pollutants were also available in this campaign (Table 1). Meteorological parameters including temperature, relative humidity (RH), wind direction (WD), wind speed (WS) and solar radiation (RS) were measured concurrently at this site.

The measurement principle and operational procedure of the TAG system have been detailed in previous studies (He et al., 2020; Kreisberg et al., 2009; Q. Wang et al., 2020; Williams et al., 2006; S. Zhu et al., 2021). In brief, the TAG system was operated with a time resolution of 2 hour, with the first hour spent on sample collection at a flow of 10 L/min and the second hour on GC-MS analysis. A total of 98 polar and nonpolar organic compounds were identified and quantified in this study and the full list is provide in Table S1. The detailed quality control measures and results for the TAG measurements have been reported in S. Zhu et al. (2021) and given in section 2.2.1.

**Table 1.** Comprehensive online instruments adopted for this campaign.

| Instrument | Parameters | Time resolution | Model (Manufacturer) |
|---|---|---|---|
| Thermal desorption Aerosol Gas chromatography-mass spectrometry system | Organic molecular markers in $PM_{2.5}$ | 1 hour | TAG (Aerodyne Research Inc., USA) |
| Monitor for AeRosols and Gases | Inorganic water-soluble ions ($NO_3^-$, $SO_4^{2-}$, $Cl^-$, $NH_4^+$, $Na^+$, $Mg^{2+}$, $Ca^{2+}$, $K^+$) in $PM_{2.5}$ | 1 hour | MARGA ADI 2080 (Applikon Analytical B.V., Switzerland) |
| Semi-continuous OC/EC analyzer | OC, EC in $PM_{2.5}$ | 1 hour | Model RT-4 (Sunset Laboratory, USA) |
| Online non-destructive X-ray fluorescence spectrometer (XRF) | 15 trace elements (K, Ca, V, Cr, Mn, Fe, Ni, Cu, Zn, As, Se, Ba, Pb, Si, and S) in $PM_{2.5}$ | 1 hour | Xact® 625 Ambient Continuous Multi-metals Monitor (Cooper Environmental Services, USA) |
| Online beta attenuation particulate monitor | $PM_{2.5}$ | 1 min | FH 62 C14 series (Thermo Fisher Scientific Inc., USA) |
| $NO_x$ monitor | NO, $NO_2$ | 1 min | Model 42i (Thermo Fisher Scientific Inc., USA) |
| $O_3$ monitor | $O_3$ | 1 min | Model EC9811 (Ecotech Inc., Australia) |
| Online gas chromatography systems equipped with flame ionization detector (GC‐FID) | $C_2$ - $C_{12}$ VOCs | 30min | Chromato-sud airmoVOC C2-C6 #5250308 and airmoVOC C6-C12 #2260308, (Chromatotec, Bordeaux, France) |
| Aerosol mass spectrometer | Organics in $PM_1$ | 1 min | AMS (Aerodyne Research Inc., USA) |

### 2.2 Data analysis

### 2.2.1 Data quality and control

A total of 638 valid samples were measured by TAG throughout the field campaign. A mixture of 20 deuterated compounds was added as internal standards in analysis of each sample and in calibration, to track and correct the changes in instrumental





sensitivity. Table S1 lists the range and average concentrations of the 98 quantified organic compounds, together with their respective quantification ions and internal standards. For the ease of discussion, the 98 TAG-measured organic compounds are sorted into 18 compound groups in the following discussions, labeled as alkanes, hopanes, polycyclic aromatic hydrocarbons (PAHs), primary sugars (PSs), biomass burning tracers (BBtracers), unsaturated fatty acids (uFAs), saturated fatty acids (sFAs), aromatic polycarboxylic acids (Ar-PCAs), nitro-aromatic compounds (NACs), $C_9$ acids, $C_{6-8}$ hydroxyl dicarboxylic acids (H_hDCAs), $C_{6-8}$ dicarboxylic acids (H_DCAs), $C_{3-5}$ hydroxyl dicarboxylic acids (L_hDCAs), $C_{3-5}$ dicarboxylic acids (L_DCAs), phthalic acid (Pht), 2,3-dihydroxy-4-oxopentanoic acid (DHOPA), β-caryophyllene tracers (βCaryT) and α-pinene tracers (αPinT), considering both compound structures and commonality in source origins. Molecules lumped into the same group normally show correlations with each other with $R_p$ higher than 0.6 (Figure S1). We further evaluated the quality of hourly dataset by conducting multiple cross-comparisons among independent measurements, of which scatter correlation plots are illustrated in Figure S2. For example, the summed mass of 98 TAG-measured organic molecules is well correlated with OC measured by OC/EC analyzer ($R^2$ = 0.73) and total organics measured by AMS ($R^2$ = 0.74). TAG-measured benzo[ghi]perylene correlated well with EC measured by OC/EC analyzer ($R^2$ = 0.72), which is consistent with residual oil combustion as their dominant common source. TAG-measured hopanes and fatty acids are well-correlated with the hydrocarbon-like OA (HOA, $R^2$ = 0.60) and cooking OA (COA, $R^2$ = 0.74) resolved from the mass spectra by AMS, respectively, reflecting vehicular emissions (VE) and cooking activities as their respective common source. And those secondary organic molecules (e.g., pinic acid, DHOPA, phthalic acid, DCAs, and hDCAs) measured by TAG showed moderate to strong correlations ($R^2$ = 0.21~0.68) with nitrate, sulfate measured by MARGA and secondary source factors (e.g., MO-OOA, LO-OOA) derived from AMS. Besides, the inorganic ions ($NO_3^-$, $SO_4^{2-}$, $NH_4^+$) measured by AMS fairly well correlated with those measured by MARGA ($R^2$ = 0.59~0.79). The summed secondary source factors derived from AMS also showed strong correlations ($R^2$ = 0.87) with SOM estimated by OC/EC ratio method, and its summed primary source factors correlated well with estimated POM ($R^2$ = 0.44). Overall, the data consistency checks indicate that the TAG system and other online instruments have provided good quality measurements.

### 2.2.2 Estimation of primary and secondary organic aerosol mass concentrations

OC in the ambient $PM_{2.5}$ can be apportioned into primary OC (POC) and secondary OC (SOC) according to their source origins. As it is analytically infeasible for direct measurement of POC and SOC, an estimation method based on OC/EC ratio is widely adopted (Castro et al., 1999; Turpin and Huntzicker, 1995). Specifically, EC serves as a tracer to track the portion of co-emitted POC and the following equations are used to calculate POC and SOC:

$$POC = EC \times (OC/EC)_{pri} \tag{1}$$

$$SOC = OC - POC \tag{2}$$

where OC and EC are the measured hourly concentrations of OC and EC, and $(OC/EC)_{pri}$ is the average OC-to-EC ratio from primary emission sources. In this study, the minimum OC/EC ratio of 1.5 during the campaign was adopted to represent $(OC/EC)_{pri}$ (Lim and Turpin, 2002). This value fell in the range reported in other studies (1.4~1.9) for estimating POC and SOC concentrations in Shanghai (D. Wang et al., 2016; Yao et al., 2020; Zhao et al., 2015). Subsequently, primary organic matter (POM) and secondary organic matter (SOM) were calculated from POC and SOC by multiplying conversion factors of 1.4 and 2.0, respectively. The multipliers were previously reported in W. Zhu et al. (2021) based on the 2016-2017 AMS data measured at the same site.

### 2.3 Clustering analysis and concentration weighted trajectory (CWT)



Backward trajectories for air masses arriving at the observation site and their clustering analysis were calculated every hour by HYSPLIT software (http://ready.arl.noaa.gov/HYSPLIT.php) with 6-hour archived GDAS (Global Data Assimilation System) data. Based on the change in total spatial dissimilarity (TSV) (Figure S3) and variations of $PM_{2.5}$ chemical composition under each cluster (Figure S4), an optimal solution of four clusters (Figure S4), representing marine, local, YRD transported and long-range transported air masses, was determined. The information derived from HYSPLIT was then used to determine the potential source areas for $PM_{2.5}$ under the influence of different air mass clusters and the results are illustrated by concentration weighted trajectory (CWT) approach with the adoption of ZeFir software (Petit et al., 2017). More detailed description of these analyses is given in Text S1.

## 3. Results and Discussion

### 3.1 General descriptions

Figure 1 shows the time series of meteorological parameters, gaseous pollutants, and $PM_{2.5}$ and its chemical components during the campaign. The average $PM_{2.5}$ mass loading was $49.9 \pm 36.9$, $\mu g/m^3$ and significant hour-to-hour variation was recorded. $PM_{2.5}$ episodes were identified to be periods of hourly concentrations exceeding 35 $\mu g/m^3$ and durations over 20 hours. A total of nine $PM_{2.5}$ episodes (EP#1 – EP#9) thus emerged throughout the study period and are individually labeled in Figure 1. Among them, EP#1, EP#7 and EP#8, which lasted from 31 to 105 hours, were categorized into "transport episodes" on the basis that their trajectories with high particle concentrations originated from Shandong province and passed over YRD region before reaching Shanghai (Figure 2a). EP#3, EP#4 and EP#6 were categorized as "local episodes", as they were characterized by significantly lower moving speed of polluted air parcels circling around Shanghai (Figure 2a). Compared with transport episodes, the durations of local episodes were normally much shorter, ranging from 21 to 38 hours. EP#2, #5 and #9, each lasting over 4 days with high $PM_{2.5}$ concentrations originated from both the YRD region and local areas (Figure 2a), were thus termed as "mixed-influence episodes". The remainder days were classified as non-episodic periods, characterized by notably lower concentrations of most ambient pollutants (e.g., $PM_{2.5}$, $NO_x$, VOCs). Consistent with previous studies (M. Li et al., 2019; Y. Wang et al., 2014b; Wei et al., 2019), the occurrences of haze episodes in Shanghai during wintertime were associated with air parcels originating from the YRD region or local areas under stagnant meteorological conditions, while the clean periods were characterized by prevailing air masses that were transported long-range or of marine origin and were associated with higher wind speed (WS > 4 m/s) which favored the diffusion and dilution of air pollutants. More detailed statistics related to the average values of meteorological parameters, ambient pollutants, $PM_{2.5}$ major components and diagnostic ratios during individual episode categories and non-episodic periods are summarized in Table 2.

Among the three types of episodes, $PM_{2.5}$ concentration showed the highest average level during transport episodes ($83.5 \pm 37.0 \, \mu g/m^3$) with hourly concentration fluctuating from 32 to 178 $\mu g/m^3$, followed by the mixed-influence episodes ($78.0 \pm 29.5 \, \mu g/m^3$) and local episodes ($62.4 \pm 25.3 \, \mu g/m^3$) (Table 2). During the transport and mixed-influence episodes, high concentrations of $PM_{2.5}$ were observed along with relatively higher concentration of $O_3$ under lower level of RH (< 70%) and higher intensity of solar radiance (RS > 80 $W/m^2$). Local episodes generally occurred with a notable drop of WS ($2.3 \pm 1.4$ m/s) and relatively higher level of RH ($83.7 \pm 9.3\%$). Apparently, the stagnant meteorological conditions were favorable for the accumulation of pollutants from local emissions. Significant higher levels of $NO_x$ ($98.2 \pm 46.6$ ppbv) and volatile organic compounds (VOCs) ($74.5 \pm 31.5$ ppbv), as well as $NO/NO_2$ ($1.30 \pm 1.09$) and toluene/benzene (T/B, $3.8 \pm 1.7$) ratios, were also observed during local episodes, reflecting their origin of local vehicular and combustion sources with less influence from aging processes.



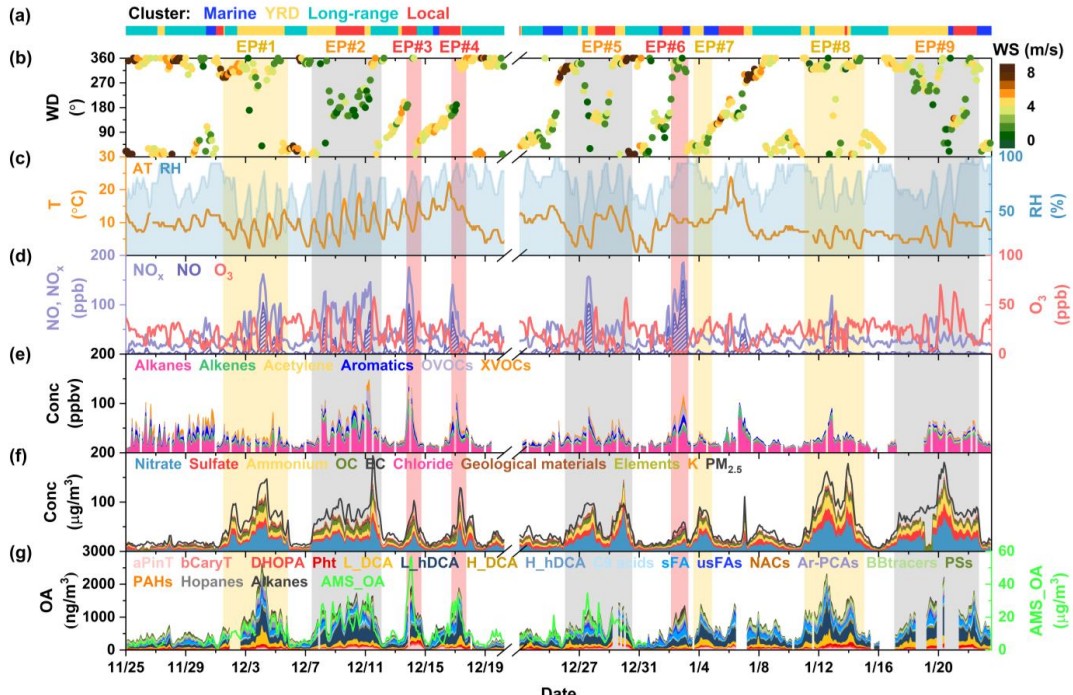

**Figure 1.** Time series of (a) air mass clusters; (b) wind direction (WD), wind speed (WS); (c) temperature (T), relative humidity (RH); (d) NO, $NO_x$, $O_3$; (e) VOCs measured by GC-MS (see Table S2 for the VOC groups and individual VOCs); (f) major components of $PM_{2.5}$ and total $PM_{2.5}$ (dark line); and (g) organic molecular groups in $PM_{2.5}$ measured by TAG and total OA (green line) in $PM_1$ measured by AMS during the campaign. The concentration of geological materials was calculated to be 2.49[Si] + 1.63[Ca] + 2.42[Fe], and concentration of elements was calculated as the sum of V, Cr, Mn, Ni, Cu, Zn, As, Se, Ba, and Pb. The nine episodic events were shaded in yellow, grey and red, representing transport, mixed-influence and local episodes, respectively.

**Table 2.** Summary of meteorological parameters, ambient pollutants, $PM_{2.5}$ major components and diagnostic ratios for different types of episodes and non-episodic periods.

| Parameters | Local episodes | Mixed-influence episodes | Transport episodes | Non-episodic periods |
|---|---|---|---|---|
| | EP#3, #4, #6 | EP#2, #5, #9 | EP#1, #7, #8 | / |
| *Meteorological factors* | | | | |
| RH (%) | 83.7 ± 9.3 | 69.2 ± 17.8 | 67.8 ± 17.4 | 76.8 ± 13.7 |
| WS (m/s) | 2.3 ± 1.4 | 3.1 ± 2.1 | 3.4 ± 1.7 | 4.4 ± 2.0 |
| RS (W/m²) | 41.0 ± 110.2 | 95.9 ± 167.4 | 86.0 ± 151.9 | 60.7 ± 125.3 |
| *Ambient pollutants* | | | | |
| $PM_{2.5}$ (µg/m³) | 62.4 ± 25.3 | 78.0 ± 29.5 | 83.5 ± 37.0 | 22.6 ± 12.2 |
| $NO_x$ (ppbv) | 98.2 ± 46.6 | 48.7 ± 32.2 | 46.3 ± 30.1 | 29.6 ± 14.1 |
| $O_3$ (ppbv) | 5.5 ± 7.2 | 21.5 ± 15.9 | 19.7 ± 11.5 | 20.9 ± 9.6 |
| VOCs (ppbv) | 74.5 ± 31.5 | 48.8 ± 24.6 | 30.3 ± 13.8 | 27.8 ± 17.8 |
| *$PM_{2.5}$ composition* | | | | |
| Nitrate (%) | 24.8 ± 6.7 | 33.1 ± 8.6 | 34.6 ± 6.6 | 21.2 ± 7.8 |
| Sulfate (%) | 10.4 ± 2.5 | 14.8 ± 6.0 | 12.3 ± 3.8 | 17.7 ± 5.9 |
| Ammonium (%) | 11.7 ± 2.2 | 15.3 ± 3.6 | 14.8 ± 2.2 | 12.4 ± 3.4 |
| SOM (%) | 26.5 ± 10.6 | 15.8 ± 5.4 | 13.3 ± 3.8 | 20.5 ± 8.3 |
| POM (%) | 6.6 ± 2.8 | 3.1 ± 1.1 | 3.4 ± 1.0 | 5.4 ± 2.5 |



| | | | | |
|---|---|---|---|---|
| Others (%) | $20.0 \pm 13.3$ | $17.9 \pm 15.0$ | $21.6 \pm 9.2$ | $22.8 \pm 15.3$ |
| *Ratios* | | | | |
| $NO/NO_2$ | $1.30 \pm 1.09$ | $0.26 \pm 0.36$ | $0.26 \pm 0.35$ | $0.19 \pm 0.21$ |
| $NO_3^-/SO_4^{2-}$ | $2.55 \pm 1.00$ | $2.60 \pm 1.12$ | $3.07 \pm 1.00$ | $1.31 \pm 0.63$ |
| Toluene/Benzene | $3.8 \pm 1.7$ | $1.8 \pm 1.1$ | $1.7 \pm 1.2$ | $2.2 \pm 1.6$ |

180    Figure 2 (b) shows chemical compositions in $PM_{2.5}$ and mass percentages of secondary organic matters (SOM) during the nine episodes as well as non-episodic periods, and Figure 2 (c) compares the mass increment ratios and mass percentages of SOM with that of combined secondary inorganic aerosols (SIA) among different episodes. In general, secondary species (e.g., $NO_3^-$, $SO_4^{2-}$, $NH_4^+$, SOM) constituted the largest fraction of $PM_{2.5}$ during both polluted (68%-86%) and clean (75%) periods, yet the composition was substantially different. Mass contributions of secondary inorganic aerosol (SIA, including $NO_3^-$, $SO_4^{2-}$

185    and $NH_4^+$) to $PM_{2.5}$ were much higher during transport episodes and mixed-influence episodes. Especially for nitrate, which accounted for 31%-40% of $PM_{2.5}$ average concentration during transport episodes versus 23%-28% of $PM_{2.5}$ mass concentration during local episodes and non-episodic days. In contrast, SOM took up a more prominent portion in $PM_{2.5}$ during local episodes ranging from 22% to 27%. The highest portion of SOM (27%) occurred during the local episode EP#6, and this fraction even exceeded nitrate (26%).

190    While SIA had comparable percentage contributions to $PM_{2.5}$ during all episodes (46%-72%), higher mass incrementation of SOM was observed during local haze episodes with a ratio of 2.8-3.9 to non-episodic periods, highlighting the importance of secondary organic aerosol formation in local $PM_{2.5}$ pollution. Indeed, primary species (e.g., POM, EC, potassium, chloride, geological material matters and other trace elements) also showed noticeable increases with summed contributions up to 29% during local episodes, while their percent contributions during the transport and mixed-influence episodes were in the range

195    of 8-14%. The higher proportions of primary species together with significantly higher values of $NO/NO_2$ and T/B ratios indicate that local $PM_{2.5}$ episodes in Shanghai were largely influenced by freshly emitted primary pollutants in the local areas. These results suggest largely different sources and chemical processing of $PM_{2.5}$ formation under different haze types.

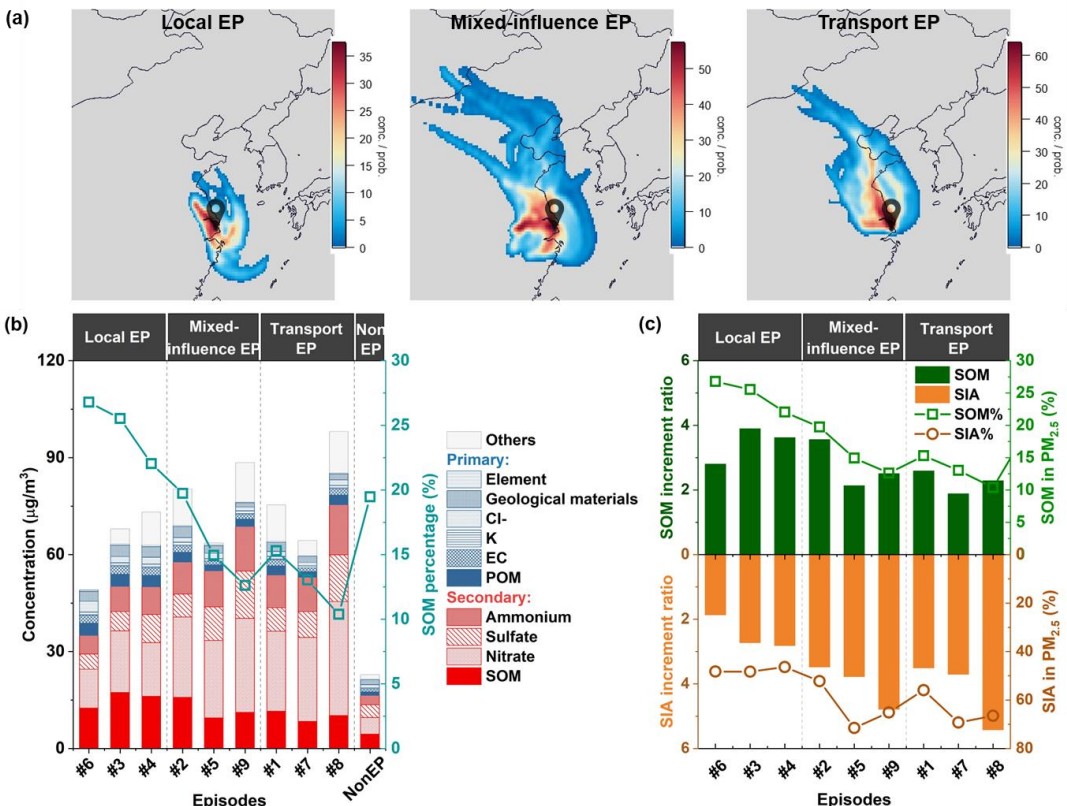

**Figure 2.** (a) Concentration weighted trajectory (CWT) maps for PM$_{2.5}$ (the droplet icon in the maps represents the location of the observation site) and (b) its chemical compositions during different episodes; (c) comparisons of mass increment ratios of SOM and combined SIA during different episodes in reference to non-episodic periods. Mass increment ratio close to 1 indicates no obvious increment.

### 3.2 Characteristics of organic compound groups during haze episodes

#### 3.2.1 Major classes of organic compounds in PM$_{2.5}$

The average concentration of total 98 organic compounds measured by TAG system during the campaign was 809 ± 499 ng/m$^3$. Among the quantified OA markers, the L_hDCAs group exhibited the highest concentration (264 ± 187 ng/m$^3$), which was dozens to hundreds of times higher than those of the other groups. Malic acid and glyceric acid were the main components of L_hDCAs, the former of which was also the most abundant individual compound among all 98 measured compounds. The average concentrations of malic acid and glyceric acid during the campaign were 156 ± 112 ng/m$^3$ and 54 ± 44 ng/m$^3$, respectively. These levels were at the same magnitude as those observed at urban sites in Hong Kong (Hu et al., 2008; Hu et al., 2013; Lyu et al., 2020). The concentration level of L_DCAs was only second to that of L_hDCAs with an average value of 95 ± 83 ng/m$^3$. The high mass concentrations and proportions of these highly oxidized organic molecules (L_hDCAs and L_DCAs) indicates that aerosols measured at this site were frequently aged. Of comparable concentration to L_DCAs was saturated fatty acids (sFAs) (93 ± 80 ng/m$^3$), signaling the influence of cooking activities on PM$_{2.5}$ at this urban site. As listed in Table S1, BBtracers, which are specific organic molecular tracers for biomass burning, had an abundance level of (72 ± 41





ng/m$^3$). Ar-PCAs were indicators for secondary products of biomass burning emissions (He et al., 2018; J. Schauer et al., 2002). The sum concentration of Ar-PCAs was $40 \pm 33$ ng/m$^3$ during the campaign. Ar-PCAs were well correlated with both BBtracers (Figure S1) and secondary inorganic ions (Figure S5). The relatively high concentrations of BBtracers and Ar-PCAs among the 18 groups implies that biomass burning activities during wintertime still persisted and transported to urban Shanghai despite

the prohibition of field fires implemented in recent years.

### 3.2.2 Comparison of OA variations between local and transport episodes

Table 3 reveals distinct concentration levels of organic markers for different air pollution types and Figure S6 shows their mass percentages. In general, the proportions of organic molecular groups in PM$_{2.5}$ differed among different episodic types. During local episodes, the TAG-measured OA (average 1409 ng/m$^3$) was characterized by sizable contributions from primary

and secondary anthropogenic organic molecules, including alkanes, PAHs, hopanes, uFAs, sFAs, C9 acids, DHOPA, pht, which contributed 47% of the mean mass concentration. On the contrary, the TAG-measured OA during transport episodes (average: 1164 ng/m$^3$) was dominated by secondary organic molecular groups. L_DCAs ($167 \pm 103$ ng/m$^3$) and L_hDCAs ($358 \pm 176$ ng/m$^3$). These two groups have been highlighted as later oxidation products of primary precursors with a wide range of sources, and they made up 45% of the mean mass concentration. Despite the notably lower contributions (32%) from the sum of primary

organic molecules (sFAs, uFAs, BBtracers, PSs, alkanes, PAHs and hopanes) during transport episodes, the average proportion of BBtracers in TAG-measured OA ascended from 5% during local episodes to 10% during transport episodes. Higher contributions from Ar-PACs and NACs were also observed during the mixed-influence and the transport episodes compared with the local episodes, suggesting that biomass burning played an important role in the accumulation of transported PM$_{2.5}$. Overall, TAG-measured OA was over 20% more abundant in mass concentration in the atmosphere during local episodes

compared to the mixed-influence episodes and the transport episodes.

**Table 3.** Mean Levels of TAG-measured organic molecular groups and total OA during different types of PM$_{2.5}$ pollution episodes.

| Organic molecular groups* (ng/m$^3$) | Local episodes EP#3, #4, #6 | Mixed-influence episodes EP#2, #5, #9 | Transport episodes EP#1, #7, #8 | Non-episodic periods / |
|---|---|---|---|---|
| αPinT | $75.9 \pm 24.9$ | $41.6 \pm 26.1$ | $39.3 \pm 18.9$ | $15.4 \pm 15.3$ |
| βCaryT | $2.97 \pm 0.85$ | $1.68 \pm 0.83$ | $2.12 \pm 1.03$ | $0.79 \pm 0.80$ |
| DHOPA | $45.6 \pm 12.6$ | $20.9 \pm 11.3$ | $17.9 \pm 11.3$ | $5.2 \pm 4.9$ |
| Pht | $61.0 \pm 22.4$ | $32.0 \pm 20.2$ | $48.4 \pm 24.3$ | $17.7 \pm 14.2$ |
| L_DCAs | $96.4 \pm 46.4$ | $154.6 \pm 70.7$ | $166.6 \pm 102.7$ | $42.5 \pm 29.3$ |
| L_hDCAs | $304.8 \pm 120.4$ | $406.4 \pm 197.2$ | $357.7 \pm 176.1$ | $157.5 \pm 108.7$ |
| H_DCAs | $43.3 \pm 18.3$ | $25.2 \pm 12.7$ | $19.3 \pm 12.4$ | $7.6 \pm 5.6$ |
| H_hDCAs | $57.0 \pm 22.6$ | $40.6 \pm 22.8$ | $37.1 \pm 23.5$ | $11.4 \pm 11.7$ |
| C9 acids | $40.9 \pm 16.5$ | $27.4 \pm 18.3$ | $17.6 \pm 10.3$ | $9.6 \pm 7.6$ |
| sFAs | $292.3 \pm 145.9$ | $110.2 \pm 59.7$ | $109.7 \pm 57.1$ | $60.0 \pm 47.4$ |
| uFAs | $101.3 \pm 81.8$ | $35.4 \pm 32.3$ | $28.0 \pm 26.2$ | $18.5 \pm 21.1$ |
| Ar-PCAs | $39.2 \pm 15.3$ | $53.3 \pm 30.9$ | $77.7 \pm 40.0$ | $22.2 \pm 16.1$ |
| NACs | $6.14 \pm 3.06$ | $10.3 \pm 7.12$ | $9.17 \pm 5.39$ | $2.49 \pm 2.51$ |
| BBtracers | $73.3 \pm 28.1$ | $81.8 \pm 33.5$ | $118.4 \pm 40.4$ | $51.3 \pm 28.9$ |
| PSs | $42.9 \pm 15.0$ | $42.1 \pm 21.6$ | $58.8 \pm 36.9$ | $25.8 \pm 18.8$ |
| Alkanes | $110.8 \pm 47.5$ | $44.6 \pm 25.5$ | $47.7 \pm 25.2$ | $24.3 \pm 17.5$ |
| PAHs | $12.5 \pm 7.19$ | $6.01 \pm 2.61$ | $7.33 \pm 3.55$ | $2.89 \pm 1.65$ |
| Hopanes | $2.81 \pm 1.15$ | $1.34 \pm 0.97$ | $1.34 \pm 0.88$ | $0.73 \pm 0.46$ |
| TAG-measured OA | $1409.0 \pm 389.5$ | $1135.5 \pm 424.0$ | $1164.1 \pm 469.4$ | $476.1 \pm 241.9$ |

\* Full names of the listed organic molecular groups and their included compounds can be found in section 2.2.1 and Table S1.

Figure 3 compares the mass abundances of four groups of species between episodic periods and non-episodic periods, including (1) individual TAG-measured organic molecules in total TAG-measured OA, (2) major components in PM$_{2.5}$, (3)



subgroups of OA in PM$_1$ as resolved by PMF analysis of the AMS data, and (4) single VOC out of the sum of a select set of VOCs, OVOCs, and XVOCs, in which OVOCs denotes oxygenated volatile organic compounds and XVOCs refers to halogenated VOCs. Species positioned above the 1:1 line indicate enhancement during episodes. The results show that different sets of species were enhanced during the three episode types. During local episodes, the OA mass increments were mainly attributable to those from vehicular emissions (e.g., hopanes) and cooking activities (e.g, fatty acids). The mass percentages

of primary vehicular-emitted tracers, such as PAHs, alkanes and hopanes increased from 0.6%, 5.1%, and 0.1% in total TAG-measured OA during clean periods to 0.9%, 7.9% and 0.2% during local episodes, respectively. Saturated and unsaturated fatty acids showed a drastic increase from 16% (non-episodes) to 28% (local episodes). Other inorganic species, including EC, chloride, and elements also exhibited higher mass proportions in PM$_{2.5}$ during local episodes, indicating that local primary emissions such as vehicle exhaust and cooking played an important role in the formation of hazes. In addition to these primary

components, vehicular and industrial source-related secondary compounds, such as DHOPA and phthalic acid also showed elevated contributions in the total TAG-measured OA with percentages of 3.2% and 4.3%, respectively, during local episodes. These results suggest that local anthropogenic sources were major contributors to elevating PM$_{2.5}$ pollution. Note that unlike sFAs and uFAs, the C9 acids, which were mainly ozone oxidation products of uFAs, did not show drastic increase during local episodes in their mass proportions in TAG-measured OA and AMS-derived OCOA in PM$_1$. This could be explained by the

significantly lower O$_3$ concentration during local episodes in comparison with the non-episodic periods (Table 2).

Different from local episodes, the majority of primary components during mixed-influence and transport episodes decreased in mass percentage compared with non-episodic periods. The mass percentages of sFAs, uFAs, alkanes, and hopanes in total TAG-measured OA decreased to 9.4%, 2.4%, 4.1% and 0.11% during transport episodes from 12.6%, 3.9%, 5.1%, and 0.15% during clean periods, respectively. Exceptions were BBtracers, PSs, and PAHs, which exhibited comparable proportions

during transport episodes and non-episodic periods. Similarly, primary inorganic species and AMS-derived primary OAs in PM$_1$ also decreased in mass abundance during the mixed-influence and transport episodes. On the other hand, mass proportions of most secondary organic molecular groups elevated, with the summed values reaching 68% during transport episodes and 72% during mixed-influence episodes, which were remarkably higher than 55% during local episodes and 61% during non episodic periods. Noticeably, those TAG-measured organic molecules that have increased in mass percentages during local

episodes were generally less-oxidized compared with that during mixed-influence and transport episodes, in consistent with the observation that transported PM$_1$ contained higher proportions of more-oxidized organic aerosol (MO-OOA) while less-oxidized organic aerosol (LO-OOA) accounted for more PM$_1$ mass during local episodes. This suggests that aerosols during the mixed-influence and transport episodes were generally more aged than local episodes.



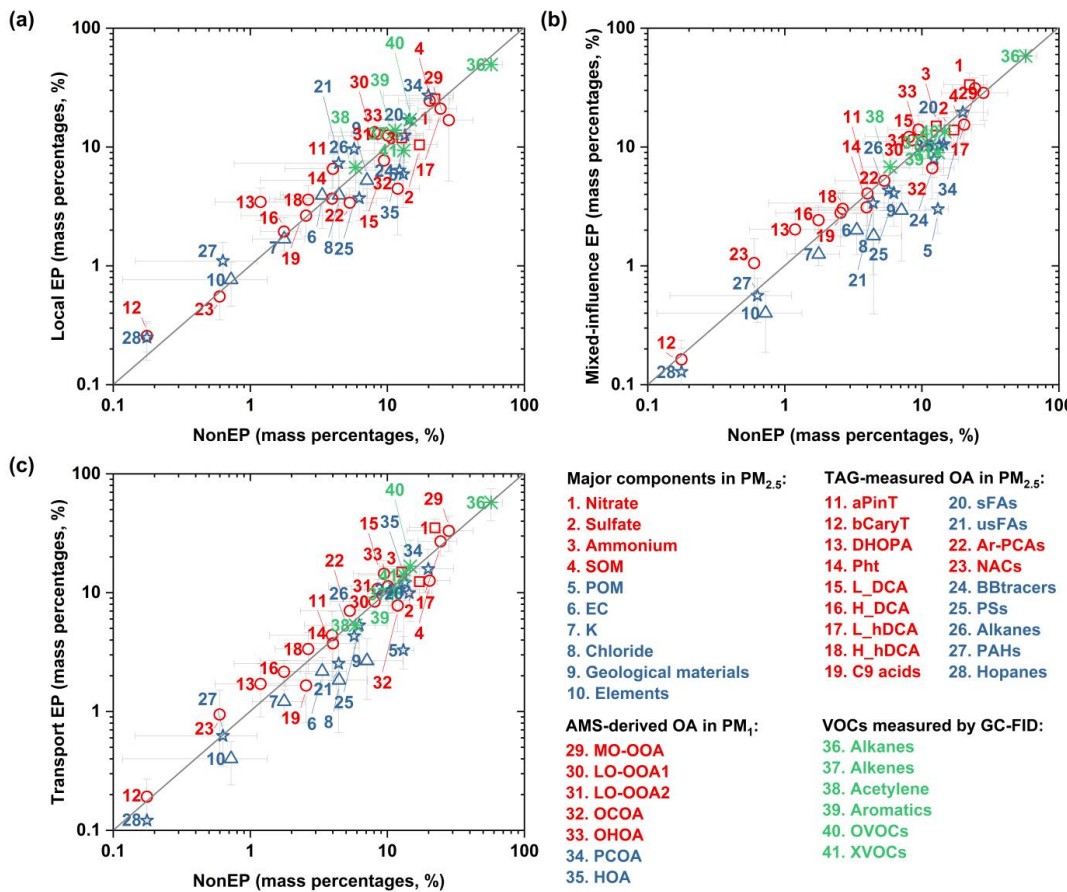


**Figure 3.** Comparison of measured VOCs and PM components in mass percentages between (a) local episodes, (b) mixed-influence episodes, and (c) transport episodes against non-episode periods. The comparison plots cover four groups of mass percentages, namely, individual organic molecule in total TAG-measured OA, major components in $PM_{2.5}$, sub-categories of bulk OA in $PM_1$, and single VOC in the sum of a select group of VOCs. Data are grouped by colors and symbols, with red open circles representing secondary origins, blue open triangles representing primary origins, green asterisks donating VOCs (e.g., alkanes, alkenes, acetylene, aromatics), OVOCs, and XVOCs in total VOCs. The measured VOC species included in alkanes, alkenes, acetylene, aromatics, OVOCs are given in Table S2. Detail information related to the identification and quantification of AMS-derived OA subgroups in $PM_1$ can be referred to Huang et al. (2021). Data located above the 1:1 line indicate an increase in respective mass proportions during episodes as compared with non-episodic periods.

## 3.3 Variations of secondary organic molecular tracers during episodes

### 3.3.1 2,3-dihydroxy-4-oxopentanoic acid (DHOPA) and aromatic SOA estimates

As discussed in the previous sections, DHOPA in TAG-measured OA had remarkable elevation in both absolute mass concentration and mass proportion during local episodes. Correlation analyses of DHOPA versus other source tracers during different episodes were performed and shown in Figure S7. The moderate to strong correlations between DHOPA and estimated





SOM during all nine episodes reaffirmed the secondary nature of DHOPA. DHOPA played a larger role in SOA formation
during local episodes in comparison with the mixed-influence and transport episodes, as suggested by the generally higher $R^2$
(0.55~0.75) and slopes (1.3~1.6) during local episodes.

It is also informative to examine the correlations of DHOPA with primary tracers. DHOPA had strong correlations with
hopanes during local episodes ($R^2$: 0.62 to 0.82), but the correlations were much weaker during transport episodes ($R^2$:
0.34~0.56) and nearly negligible during mixed-influence episodes ($R^2$: 0.00~0.28). In contrast, DHOPA had stronger
correlations with PAHs (e.g., BbF, BkF, BaF, BeP, BaP, IcdP, BghiP) ($R^2$: 0.53~0.72) during transport episodes, which was
likely related to coal combustion. Such differences could be explained as a result of differing aromatic precursor sources for
DHOPA among local versus transport episodes, with the dominating precursor sources being the vehicular emissions during
local episodes versus sources such as coal combustion and biomass burning under the influence of transported air.

Taking advantage of the DHOPA data, we used a modified tracer-based method proposed by Gao et al. (2019) and Zhang
et al. (2021a) to estimate aromatic SOA from ambient DHOPA measurements with gas-particle partitioning effects taken into
consideration. The aromatic SOA could be viewed to consist of (1) semi-volatile aromatic SOA (SemiASOA) which is formed
via gas-particle partitioning processes, and (2) more-oxidized aromatic SOA (MoASOA) that is associated with later generation
products (e.g., oligomers and dicarbonyl compounds). Although a number of monoaromatics can form DHOPA, only toluene
and xylenes were included in the SemiASOA estimation due to their predominant presence in urban area. The well estimated
hourly DHOPA values further confirmed this. The mass yield coefficients of toluene and xylenes under high NOx conditions
were adopted from previous chamber experiments (Al-Naiema et al., 2020) and more details about this estimation method are
provided in Text S2.

In general, a significant fraction (62%) of DHOPA was oxidized from m/p-xylenes through high-$NO_x$ pathways during
wintertime in Shanghai, regardless of episodic or non-episodic periods. Toluene only accounted for 38% of DHOPA mass
concentration in average under high $NO_x$ conditions (Figure S10). Several previous studies have also verified that it is incorrect
to attribute all DHOPA-based aromatic SOA estimation to toluene and xylenes can be a more predominant precursor in
aromatic SOA formation (Al-Naiema et al., 2020; Ma et al., 2018; Zhang et al., 2021b).

Comparing the contributions to total SOA from DHOPA-based semi-volatile aromatic SOA (SemiASOA), more-oxidized
aromatic SOA (MoASOA) and SOA produced from precursors other than monoaromatic hydrocarbons (NonASOA) between
episodic events and non-episodic periods as shown in Figure 4, drastic elevation in contributions from aromatic SOA were
observed during episodic events. During non-episodic periods, aromatic SOA constituted around 21% of total SOA in
wintertime in Shanghai, while this value rose to 32%~44% during episodic events. The mass contributions from aromatic SOA
also increased from 0.94 μg/m³ during non-episodic periods to 3.3 ~ 7.5 μg/m³ during episodic events. This enhancement of
SOA formation during episodes emphasizes the importance of controlling aromatic precursors for mitigating $PM_{2.5}$ pollution
in a megacity like Shanghai. Especially during local episodes, considerable benefits with average 38% reduction in SOA can
be expected to obtain if mono-aromatic VOCs are effectively controlled.

Among the nine episodes, notably higher contributions from SemiASOA were observed in local episodes, which
constituted 17% of total SOA in average. Relatively lower contributions (7% -14%) from SemiASOA and higher fractions (24%
-32%) of oligomers and dicarbonyl compounds (MoSOA) in total SOA were found during mixed-influence and transport
episodes. This suggests that SOA formed from aromatic hydrocarbons during mixed-influence and transport episodes generally
contained more highly oxidized organic products compared with local episodes, which is in consistent with the observation as
stated in section 3.2.





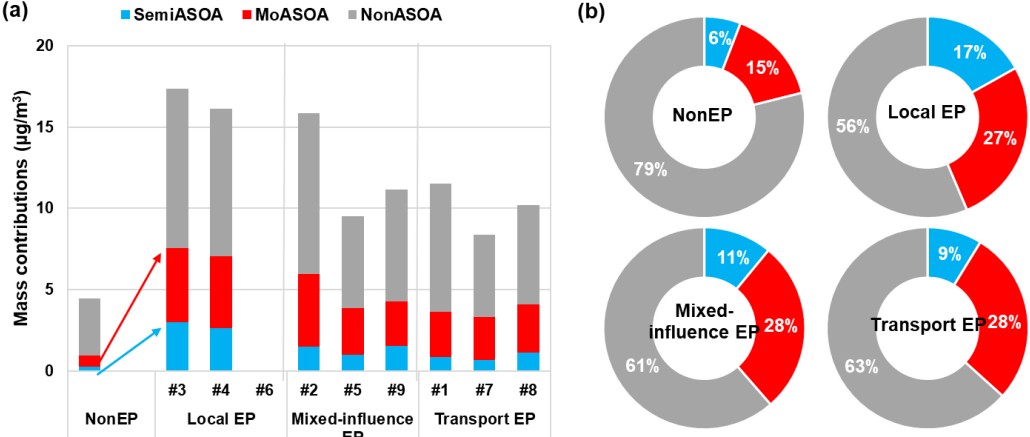

**Figure 4.** Predicted (a) mass contributions and (b) percentage contributions to total SOA from semi-volatile aromatic SOA (SemiASOA), more oxidized aromatic SOA (MoASOA) and SOA products oxidized from precursors other than aromatics (NonASOA) during episodic events and non-episodic periods. MoASOA here was calculated by subtracting SemiASOA from TotalASOA and NonASOA was estimated by subtracting TotalASOA from SOM which was calculated based on OC/EC ratio method.

### 3.3.2 Nitro-aromatic compounds (NACs)

Both primary emissions from combustion sources and nitration of aromatic hydrocarbons (e.g., benzene and toluene) are major sources of NACs in the atmosphere (X. Li et al., 2020; Y. Wang et al., 2019; Yan et al., 2017). NACs measured in this study displayed moderate to strong correlations with SOM during all episodic events (Figure S11), implying that they were likely secondarily formed from aromatic hydrocarbons. Figure 5 shows the evolution of NACs with the increase concentrations of their VOC precursors during different episodes. During the whole campaign, benzene varied in the range of 0.0-1.9 ppb while toluene varied in the range of 0.1-7.1 ppb. In general, NACs concentrations ascended with the increasing of toluene and benzene during all episodic events. During local episodes, NACs displayed a linear growth with the elevation of toluene concentrations, while they did not further increase with benzene concentrations when benzene concentrations were higher than 1.5 ppb (Figure 5a, d). In comparison, NACs exhibited a linear increasing trend with benzene concentrations during the entire concentration range of 0.0-1.9 ppb while the linear correlation of NACs with toluene ceased when toluene was higher than ~3.5 ppb during mixed-influence episodes or ~2.5 ppb during transport episodes (Figure 5e, f). Toluene is a more reactive aromatic compound and more abundantly emitted from vehicular sources. As such, it is likely that toluene was a more predominant precursor in forming NACs during local episodes. Yet during mixed-influence and transport episodes, benzene was a more dominant aromatic hydrocarbon precursor for NACs due to its relatively stable chemical structure and higher influences from coal combustion and biomass burning associated with air masses from north China.

Figure S11 further confirmed that NACs concentrations during transport episodes were largely impacted by biomass burning emissions. Hourly concentrations of NACs showed strong correlations with organic tracers indicative of biomass burning (BBtracers) with $R^2$ higher than 0.64 during transport episodes, while the correlation coefficients dropped to 0.30 during local episodes. Such results suggested that biomass burning was a major source of NACs in Shanghai during transport episodes and likely had sizeable influence during local episodes. The stronger correlations between NACs and SOM with higher values of slopes during mixed-influence and transport episodes also suggested that NACs played a more important role





in SOA formation during mixed-influence and transport episodes.

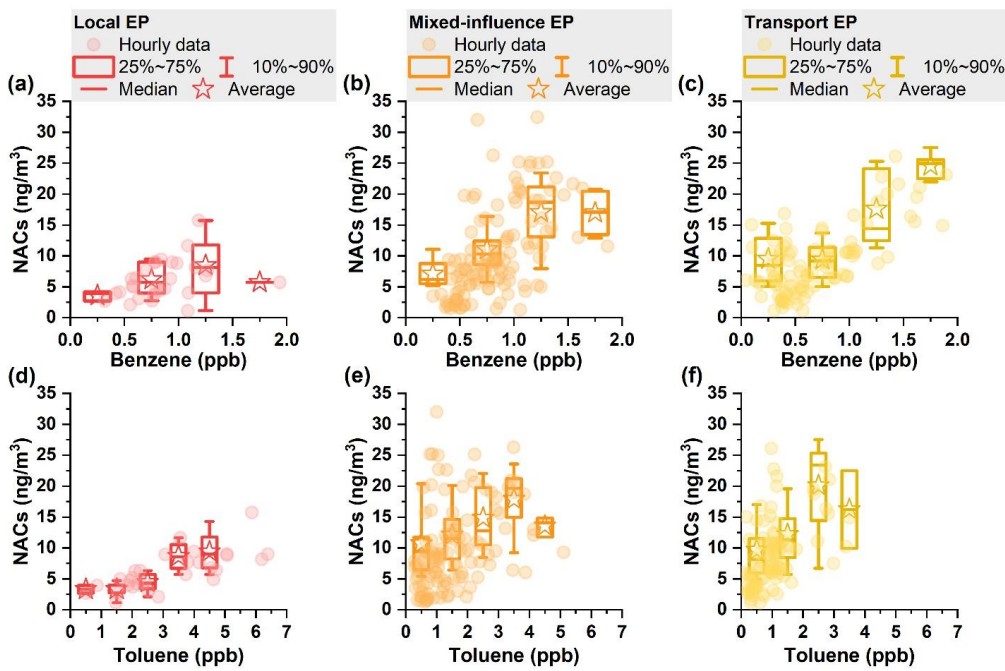

**Figure 5.** Concentrations of NACs as a function of benzene and toluene concentration bins during the three types of episodic events. The markers represent mean values and whiskers represent 25th and 75th percentiles.

Since $NO_x$ played an important role in controlling secondary products by influencing the fate of organic peroxy radicals
($RO_2$) (J. Kroll et al., 2006; J. Kroll et al., 2008; K. Nihill et al., 2021; Peng et al., 2019), correlations of $NO_3^-$/NACs ratios versus $NO/NO_2$ ratios were compared among different episodes (Figure S11). During the three local episodes, $NO_3^-$/NACs was all negatively correlated with $NO/NO_2$ ratios with $R^2$ ranging from 0.40 to 0.67. This suggested that the higher $NO/NO_2$ ratios under the influence of local air masses greatly hindered local OH level and $RO_2$ branching chemistry was dominant under this high NO environment. In contrast, no correlation was observed between $NO_3^-$/NACs and $NO/NO_2$ ratios during
mixed-influence and transport episodes, which may be attributable to the dominant role of OH reactions in both nitrate and NACs formations.

Both DHOPA and NACs are secondarily derived from mono-aromatics, with DHOPA being a benzene-ring opening product while NACs being ring-retaining products. We examined the variation pattern of the DHOPA/NACs ratio under different RH and $O_x$ level bins (Figure S12a). An evident gradient was noted as a function of RH and $O_x$ when $O_x$ concentration
level was lower than 65 ppb. Under conditions of higher RH and lower $O_x$, higher DHOPA/NACs was revealed, suggesting more conducive conditions for aqueous-phase processing in forming more-oxidized SOA. No clear trend was observed for DHOPA/NACs ratios when $O_x$ level was higher than 65 ppb. It is likely that when atmospheric oxidation capacity was enhanced, the competition between benzene-ring addition and open reactions were affected by multiple factors (e.g., abundance of VOC precursors, air masses).

Similar conclusion can be deduced from the variations of NACs versus BBtracers ratios (Figure S12b). The ratios presented the highest values on the left-top corner and experienced small changes as RH increased when $O_x$ was less than 55





ppb, indicating that gas-phase photooxidation is a more dominant pathway for the formation of NACs in winter Shanghai. Previous studies also showed that high atmospheric oxidation capacity facilitated the transformation of mono-aromatics into particle-phase NACs and increased NACs concentrations substantially (Salvador et al., 2021; Yuan et al., 2016).

### 375  3.3.3 Dicarboxylic acids and hydroxyl dicarboxylic acids (DCAs and hDCAs)

For all three types of haze episodes, L_DCAs and L_hDCAs were observed to increase significantly. Their good correlations with nitrate, sulfate, and MO-OOA reflected that they were mainly formed via secondary processes. To further provide implications for their precursor sources and aging processes, diagnostic ratios of DCAs and hDCAs during episodic and non-episodic periods are examined as a function of $O_x$ and RH in Figure 6. On the one hand, both succinic acid (C4) and

glutaric acid (C5) could be formed from a wild range of precursors of longer carbon chains, while C4 could also be the product of C5 via undergoing successive oxidation cleavage (Ervens et al., 2004; Kawamura and Bikkina, 2016; Yang et al., 2008). On the other hand, C4 can be further oxidized by OH radical to form malic acid (hC4) (Ervens et al., 2004; Yang et al., 2008). Therefore, the ratios of C4/C5 and hC4/C4 could be applied to indicate the oxidizing degree of organic aerosols and the extent of photooxidation in the atmosphere (Yu et al., 2021). An examination of episode-specific showed that the average values of

C4/C5 ratios and hC4/C4 ratios generally increased with $O_x$ level during both episodic events and non-episodic periods (Figures 6a and 6b). In addition, the hC4/C4 ratios also displayed significantly positive correlations with RH during both episodic and non-episodic periods (Figure 6e). A previous field study also observed elevated hC4/C4 ratios with the increase of RH during wet season, attributing to the aqueous-phase processing of dicarboxylic acids (Yu et al., 2021). The hC4/C4 ratios elevated more rapidly during non-episodic periods compared with episodic periods. This is likely attributable to the higher

influence from marine air masses during non-episodic periods, which is consistent with previous observations that particles from bubble bursting on the ocean was laden with high mole fraction of hydroxyl groups (Aluwihare et al., 1997; Lyu et al., 2020; Russell et al., 2011). On the contrary, the C4/C5 ratios did not show clear trend with the increase of RH (Figure 6d), indicating that their dominant chemistry was insensitive to aqueous-phase processing, consistent with our understanding that the gas-phase photochemical oxidation played a more important role in the formation of DCAs.

In urban atmospheres, long-chain fatty acids, especially $C_{16}$ and $C_{18}$ fatty acids, are dominantly sources from primary cooking emissions. Azelaic acid (C9 DCA) is a major photooxidation product of unsaturated $C_{18}$ fatty acids (He et al., 2004; Kawamura et al. 1996; Robinson et al. 2006; Rogge et al. 1991). Hence, C9 DCA/sFAs ratio reflects the oxidizing degree of cooking organic aerosols. As shown in Figure 6f, C9 FCA/sFAs ratios significantly increased with $O_x$, while RH had a minor influence on the oxidation of fatty acids. Such speculation was also supported by previous field observation and chamber

experiments that $O_3$ acted as a predominant oxidant on the degradation of fatty acids (Vesna et al., 2009; Q. Wang and Yu, 2021; Zahardis et al., 2007; Ziemann et al., 2005).



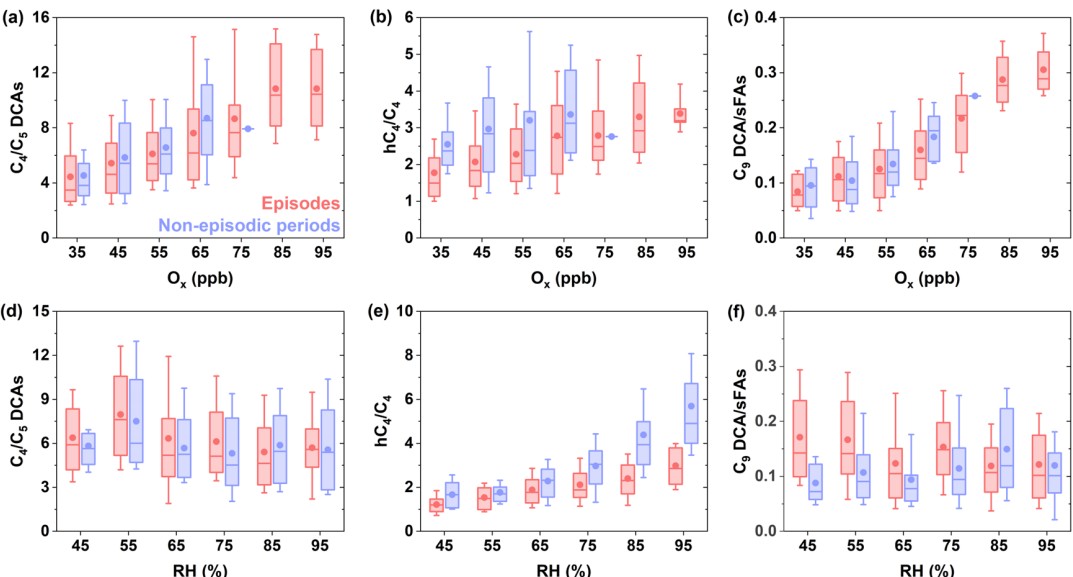

**Figure 6.** Ratios of (a) $C_4/C_5$, (b) $hC_4/C_4$, (c) $C_9/sFAs$ as a function of $O_x$ concentration bins, and (d) $C_4/C_5$, (e) $hC_4/C_4$, (f) $C_9/sFAs$ as a function of RH level bins during episodic and non-episodic events.

## 4. Conclusions

The implementation of the Air Pollution Prevention and Control Action Plan since 2013 has profoundly altered $PM_{2.5}$ chemical composition in China, with one consequence being organic aerosol constituting an increasing fraction in recent years. Yet, comprehensive understanding of the physical and chemical processing of OA has been limited. This study presents bi-hourly measurements of 98 organic molecular markers and compares their mass contributions to $PM_{2.5}$ during different types of episodes at an urban site in Shanghai, a megacity in China. The average mass concentrations of total TAG-measured OA ranged from 934 to 1595 $ng/m^3$ during the nine observed haze episodes, which were 2-3 times higher than that during non-episodic periods (476 $ng/m^3$). Enhanced OA formation was a major culprit to the deterioration of $PM_{2.5}$ pollution in wintertime in Shanghai. Major contributors of OA were substantially different among local, mixed-influence and transport episodes. Local episodes were characterized by higher contributions from primary OA markers indicative of vehicular exhaust and cooking emissions, such as alkanes, hopanes, and fatty acids, accounting for 43% of the total TAG-measured OA mass in average. The SOA markers (e.g., DHOPA, C9 acids) derived from these source categories also exhibited higher concentrations during local episodes. Specifically, the estimated mass contributions of aromatic SOA elevated from 21% during non-episodic periods to 44% during local episodes, indicating the important impacts from vehicular emissions on local aerosol formation.

In comparison, BBtracers comprised a significant contributor of primary OA during mixed-influence and transport episodes. The significant presence of BBtracers in urban $PM_{2.5}$ in Shanghai implied the continued practice of burning agricultural residuals despite recent policies of banning such activities. Consistently, Ar-PCAs and NACs, which are indicative of secondary biomass burning sources, constituted larger fractions of the total TAG-measured OA during mixed-influence and transport episodes compared with local episodes. The positive correlations between NACs/BBtracers ratios and $O_x$ during the campaign revealed that transformation of aromatics (e.g., benzene, toluene) from biomass burning via photochemical



processing wads an important source of NACs in wintertime in Shanghai. Besides, highly oxidized secondary organic molecular groups, L_DCAs and L_hDCAs, were also more abundant during mixed-influence and transport episodes under the aging of continental outflows, with contributions ranging from 39% to 57% in the total TAG-measured OA. During local episodes, L_DCAs and L_hDCAs were comparatively deficient while SemiASOA in TotalASOA were relatively higher. Such results were likely attributable to a suppression of atmospheric oxidative capacity under high $NO_x$ concentrations. The fact that L_DCAs and L_hDCAs tracked well with $O_x$ also supported their photochemical origin.

Overall, the significant variations in OA composition during different types of episodes indicate that the sources and formation processes of OA were diverse, subjecting to the influence of the prevailing air masses. Control of local urban sources such as vehicular and cooking emissions would lessen severity of local episodes while regional control of precursors for secondary inorganic aerosols and more effective restriction of biomass burning activities would reduce $PM_{2.5}$ episodes under synoptic conditions conducive for regional transport.

*Data availability.* The bi-hourly organic markers and other hourly chemical speciation data presented in this study are available from the data repository maintained by HKUST: https://doi.org/10.14711/dataset/ PXEV3I.

*Author contributions.* SZ, CH and JZY conceived the study and led the overall research. SZ made the overall data analysis with contributions from QW and SW. MZ and LQ collected and processed chemical species data measured by MARGA, OCEC analyzer and XRF. DDH collected and processed AMS measurement data. YG, SJ, QW and HW collected and processed VOCs measurement data. CC conducted background research and reviewed the writing. SZ and JZY wrote the paper with contributions from all coauthors.

*Competing interests.* The contact author has declared that none of the authors has any competing interests.

*Financial support.* This research was supported by Shanghai 2021 "Science and Technology Innovation Action Plan" social development science and technology project (21DZ1202300). We also acknowledge funding support by the Research Grants Council of Hong Kong (R6011-18) and National Natural Science Foundation of China (41875161).

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
