# Peer review of "Evolution and chemical characteristics of organic aerosols during wintertime PM2.5 episodes in Shanghai, China: Insights gained from online measurements of organic molecular markers"

_Atmospheric Chemistry and Physics, 2022_

## Author Response (AR1)

**Manuscript title: Evolution and chemical characteristics of organic aerosols during wintertime PM$_{2.5}$ episodes in Shanghai, China: Insights gained from online measurements of organic molecular markers**

**Point-by-point response to comments**

**We thank all the referees for their constructive comments. They are valuable in helping us improve our manuscript. Our point-by-point response is provided below and marked in blue text. Note the line numbers quoted in this response document correspond to the changes-tracked revised manuscript.**

**Referee #1**

**Responses to comments from Referee #1:**

1)  Can the authors explain more about why they picked 2019 winter as studying period for investigating the OA composition and evolution in Shanghai. How representative it is? Will the final conclusion of this work about the importance of control primary emissions such as cooking, vehicle, and biomass burning emissions change if the author changed the year of study?

    *Response:* The main reason that we chose 2019 winter to study OA episodic evolutions is because online measurement data from other instruments (e.g., OA in PM$_1$ from AMS, water-soluble ions in PM$_{2.5}$ from MARGA, OC and EC in PM$_{2.5}$ from OC/EC analyzer) were also available during this period, which enable us to have a more unambiguous understanding of OA formations in Shanghai. Although we did not discuss OA evolution in other years, several of our previously published papers (He et al., 2020; Huang et al., 2021; Wang et al., 2020; Zhu et al., 2021) have discussed the formation and sources of OA observed in 2018 and 2020. These studies invariably show that cooking, vehicular, and biomass burning emissions are major OA sources and have significant impacts on PM pollution in Shanghai. In this study, we take one step further to look deeper into differences of OA evolutions during different episodic events and find that local OA was more impacted by cooking and vehicle emissions while transported OA contained more biomass burning-related organic compounds. Insights learned from this work, despite the measurements conducted in 2019, will inform present and near future policymaking in improving air quality, as the general landscape of major emission sources has not changed.

2)  Section 3.1: The authors analyzed 9 episodes in total and divided them into three categories, namely, transport episodes, local episodes, and mixed-influence episodes, but in Line 154-155 the author stated that the haze episodes were under the impacts from local emission and clean episodes were influenced by long range transportation of air mass. Such claims also seem contradictory to the following statement in Line 159-161, where the author again mentioned that episodes with high average PM$_{2.5}$ level were found during transport episodes. The definitions and explanations for each types of episode should definitely be clearer in this section.

    *Response:* The transport episodes in this study refers to air masses transported from YRD region while the "long-range" in the sentence "the clean periods were characterized by prevailing air masses that were transported long-range" refers to air masses transported from much more distant areas (i.e., the north China). We have rephrased the sentence in Lines 177-179 to improve clarity.

3)  OC/EC method was used by the authors for SOM and POM estimation. Can the authors add uncertainty analysis for the SOM and POM estimations in this study? SOM as well as a variety of primary emissions were found to be dominant in the local episodes. Was the SOM partially influenced by primary emissions since primary factors were commonly strong during the local episodes? How accurate the estimation of SOM can be in this study?

    *Response:* As described in section 2.2.2, SOM and POM were estimated from SOC and POC, which in turn were calculated from EC and OC data using the EC tracer approach and assuming the (OC/EC)$_{pri}$ could be approximated by the minimum (OC/EC) value observed during this campaign. Due to its simplicity, this approach is widely adopted in the atmospheric aerosol measurement community. However, the assumptions underlying this approach dictate its large uncertainty and its imprecise nature. This deficiency is well recognized in the literature. Thus, while we agree it is desirable to quantify uncertainty for the SOM and POM estimates, currently there lacks a method for doing so. Nevertheless, we have evidence to support that the estimates are reasonable and consistent with other measurements-derived data. For example, the SOM estimated in this study exhibited strong correlations with

summed concentrations of SOA source factors derived from AMS measurements. In the revised manuscript, we have added scatter plots in Figure S3 in the revised supporting material to compare the estimated SOM and POM concentrations with source factors derived from AMS measurements to support this statement.

Regarding whether SOM was partially influenced by primary emissions, we believe the answer is yes, as the primary emissions serve as precursors for formation of SOM. The intertwining relationship of POM and SOM makes a clean separation of POM and SOM not feasible. Using PMF to separate the PM sources into primary and secondary ones is one of the indirect methods to derive at separate SOM and POM quantities. Again, the result that the SOM estimated by the EC tracer method was strongly corrected with the PMF-derived SOA factors support the robustness of the SOM estimate.

4) C9 produced from ozonolysis of fatty acids were not detected to increase in their mass content during the local episode and the author then explain this was because of the low $O_3$ mixing ratio. Did the author just mentioned the local episode was largely influenced by SOA formation based on Figure 2? Does this mean that most of SOA in the local episodes were formed form pathways other than ozonolysis, can the authors provide further evidence for this point?

*Response:* Firstly, from Table 3, Figure 3, and Figure S7 in our revised manuscript, we can see that both the absolute concentration of $C_9$ acids and their mass percentage in total TAG-measured OA increased during local episodes. However, their increases were insignificant compared with their primary precursors (i.e., fatty acids). We think the suppression of $O_3$ oxidation by the high $NO_x$ emissions is a major reason, since $NO_x$ concentrations and $NO/NO_2$ mass ratios were substantially higher during local episodes (Table 2). Additionally, the more drastic increase in the mass percentage of DHOPA during local episodes, which is a typical SOA product of monoaromatics with OH radicals, further suggested that SOA formed during local episodes were more influenced by pathways other than ozonolysis (e.g., OH oxidation). We have added more detail explanations in the revised manuscript (Line 289-296) to further clarify the variations of $C_9$ acids observed in this study.

5) From discussions of Figure 2, the authors claimed that the local episodes were significantly influenced by SOA. But in line 268, the author mentioned that AMS data indicated higher MO-OOA and LO-OOA were observed for the mixed-influence and transport episodes, indicating that OA in these two episodes were more aged. Should there at least be some assumptions given in the manuscript on how the PM1 data from AMS and the PM2.5 data from TAG were compared? The logic in terms of which episode underwent more profound secondary OA formation process is a bit mess in the current manuscript. The authors should be more explicit on which episode is more aged and has a higher formation of SOA under which specific conditions.

*Response:* The claim that local episodes were significantly influenced by SOA did not contradict the observation that SOA compositions (e.g., fresh vs. aged SOA) were different during different episodes. That is, SOA overall had higher proportions in $PM_{2.5}$ during local episodes compared with the mix-influenced and transport episodes. However, when we further investigate their OA compositions, we find that local SOA was more dominant by less-oxidized molecular groups while transported SOA contained significant higher proportions of more oxidized molecular groups (e.g., $C_{3-5}$ DCAs, $C_{3-5}$ hDCAs). The measurement data from TAG were generally consistent with those from AMS. As shown in Figure S3, the TAG-measured SOA tracers produced in early generations (e.g., DHOPA, phthalic acid, pinic acid) correlated well with LO-OOA derived from AMS while those associated with later generation products (e.g., $C_{3-5}$ DCAs, $C_{3-5}$ hDCAs) had stronger correlations with MO-OOA. The revised Figure S7 in the supporting material also confirmed that AMS-measured OA in $PM_1$ was laden with more oxidized SOA (MO-OOA) during mix-influenced and transport episodes. More explanation to reveal the diverse SOA products during different episodes are added in the revised manuscript Line 308-311.

6) The $hC_4/C_4$ ratio was used as an indicator for aqueous phase secondary OH oxidation. However, as indicated in Figure 6e, the positive correlation between $hC_4/C_4$ and RH was found to be more significant for the non-episodic period compared to the episodic period. I'm afraid the explanation of impact from marine aerosol should not be persuasive as $hC_4$ should be secondary formed according to the statement of the authors. More discussion needed to explain why the $hC_4/C_4$ ratio climbed up more significantly with the increase of RH during the non-episodic period.

*Response:* We also observed higher concentrations of $O_3$ during non-episodic periods in reference to episodic events under the same RH level bins, which appeared to support that the formation of $hC_4$ was largely facilitated by aqueous

phase OH oxidation during non-episodic periods. We have added these discussions in the revised manuscript (Lines 432-437).

7) Line 41: sentence needs to be rephrased.
   *Response:* We have rephrased the sentence in the revised manuscript (Lines 39-40).

8) Line 54-56: references needed here.
   *Response:* This sentence has been deleted as we have revised the introduction section to provide more discussions about the findings and limitations of previous field studies conducted in Shanghai (Lines 49-75).

9) Line 58-60: This sentence is hard to understand. Please provide evidence in more detail.
   *Response:* This sentence has been deleted as we have revised the introduction section to give more discussions about the findings and limitations of previous field studies conducted in Shanghai (Lines 49-75).

10) Line 67: while studying evolution processes…
    *Response:* We have rephrased the sentence in the revised manuscript (Line 81).

11) Line 105: it is very abrupt to bring up residual oil combustion here.
    *Response:* We have deleted this sentence in the revised manuscript (Lines 126-128).

12) Line 180: SOA has already been defined previously no need to have SOM here.
    *Response:* We have rephrased the sentence in the revised manuscript (Line 204).

13) Line 184: SIA has already been defined.
    *Response:* We have deleted the definition in the revised manuscript (Lines 206, 209).

14) Line 224: what does "sizable" mean.
    *Response:* "sizable" in the sentence means "fairly large" (see dictionary.com for meanings of "sizable"). Here we mean to state that those primary and secondary anthropogenic organic molecules had much higher proportions in TAG-measured OA during local episodes.

15) Line 226-227: should be local episode with high SOM contribution that have high secondary molecular tracers?
    *Response:* Yes, the local episodes with high SOM contributions also are characterized with high concentrations of secondary molecular tracers of anthropogenic origins (see Table 3). However, POM also showed higher mass concentrations during local episodes, resulting in that SOM (or SOA) did not account for a larger fraction in total OM (or total OA) during local episodes compared with mix-influenced and transport episodes.

    In Line 226-227, we meant to say, "On the contrary, the TAG-measured OA during transport episodes (average: 1164 ng/m$^3$) was dominated by secondary organic molecular groups when examining the relative proportion of primary and secondary organic molecules." We have rephrased the sentences in the revised manuscript (Lines 260-261) to improve clarity.

16) Line 296-298: I am not sure if it is appropriate to just simply define aromatic SOA compounds into these two categories---they might have overlapped zones.
    *Response:* Since we deduct semi-volatile aromatic SOA from total aromatic SOA to obtain the more-oxidized aromatic SOA, thus semi-volatile and more-oxidized aromatic SOA should have no overlapped zones. This classification has also been applied in our previously published papers (Gao et al., 2019; Zhang et al., 2021).

17) Line 397: oxidation degree
    *Response:* We have rephrased the sentence in the revised manuscript (Line 443).

**Referee #2**

**Responses to comments from Referee #2:**

1) The authors should discuss more HOMs compounds as well as their analytical analysis. Is this analytical method appropriate for their analysis? Recently offline techniques were reported to characterize several HOMs compounds in field samples including work from this group.

*Response:* In general, highly oxygenated organic molecules (HOMs) refers to a group of organic compounds which are formed in the atmosphere via autoxidation involving peroxy radicals and their chemical structures contain six or more oxygen atoms, many of which are incorporated as hydroperoxide (-OOH) functional group (Ehn et al., 2014, 2017; Bianchi et al., 2019). The -OOH functional group renders HOMs thermally liable, thus not amenable for analysis by gas chromatography (GC) techniques.

Here in our study, the majority of the organic compounds reported (e.g., DCAs, hDCAs, αPinT) contain less than six oxygen atoms. Two exceptions are 3-MBTCA (3-methyl-1,2,3-butanetricarboxylic acid, $C_8H_{12}O_6$) and mannitol ($C_6H_{14}O_6$), both containing six oxygen atoms. However, they are not regarded as HOMs since neither of them is formed via autoxidation or contains hydroperoxide functional group. Given that the organic compounds reported in this study are not HOMs and the TAG instrument, incorporating GC as part of its instrument component to achieve separation of organic mixture, is not designed for analyzing HOMs, we feel it is outside the scope of this work to discuss more about HOMs.

The reviewer mentioned that offline techniques have been developed by our group to quantify HOMs in field samples. We guess the reviewer may be referring to the papers by Nie et al. (2022) and Lu et al. (2023), which adopted CIMS and chemical-ionization orbitrap mass spectrometry to measure oxygenated organic molecules (OOMs) including several HOMs compounds. However, these instruments were not deployed during this campaign.

2) Winter pollution is associated with nitrogen-containing compounds (mainly nitrates). The authors should discuss if nitrates were observed. Are they detected using the TAG method?

*Response:* We agree with the reviewer regarding the importance of organic nitrates. Organic nitrates can play an important role in PM$_{2.5}$ pollution, especially in urban areas with high NO$_x$ emissions. However, the TAG method, which thermally desorbs the organic compounds in particle samples, is not suitable for analyzing organic nitrates due to the inherent instability of their chemical structures (the -ONO$_2$ function group). For example, peroxy nitrates (RO$_2$NO$_2$) will dissociate when temperature raises to ~150℃ and alkyl nitrates (ANs, RONO$_2$) are found to dissociate around 200~250 ℃ (Hao et al., 1994; Keehan et al., 2020). Among the nitrogen-containing compounds, only four nitro-substituted aromatic compounds (i.e., 4-nitrophenol, 4-nitrocatechol, 3-methyl-5-nitrocatechol, and 4-methyl-5-nitrocatechol) were quantified during this field campaign, as these compounds have sufficient thermal stability, with higher dissociation temperature (> 350 ℃) (Hao et al., 1994; Jaoui et al., 2018).

3) The authors report aging process occurs during these episodes based on the structure of the markers observed by the authors. The authors should discuss how they distinguished between aging OA and non-aging OA (specify the representative chemicals responsible for aging for example)? The interpretation of the markers based on the structure (more oxygenated) is presented and is a bit tedious but is important for the authors to clarify this issue.

*Response:* The unambiguous molecular information offered by the TAG system enables us to interpret OA aging processes through specific SOA tracers and their formation chemistry established in controlled chamber experiments. For example, a number of chamber studies have confirmed that pinic acid and pinonic acid are early generation SOA products of α-pinene ozonolysis while 3-MBTCA is a later generation product (kristensen et al., 2014; Ma et al., 2008; Szmigielski et al., 2007). Several studies have also shown that L_DCAs and L_hDCAs are aging SOA tracers, the formation of which require multiple oxidation steps (Ervens et al., 2004; Yang et al., 2008). Also, as shown in Figure S3 in the revised supporting material, DHOPA and pinic acid showed stronger correlations with LO-OOA derived from AMS measurements while L_DCAs and L_hDCAs showed stronger correlations with MO-OOA. This further supports our interpretations of aging and non-aging SOA. We have added more clarification in the revised manuscript (Lines 245-253).

4) The authors should clarify how the AMS and TAG DATA were used and interconnected in this study. Mainly for differentiating between the 3 categories/groups of episodes, the aging process, and SOA vs POA. They are areas in

the manuscript where these processes need to be carefully and explicitly discussed and more cautious about the reconciliation between the two methods AMS and TAG (PM1 and PM2.5 analyzed by AMS and TAG respectively).

*Response:* The measurement data from TAG were generally consistent with those from AMS. As shown in Figure S3 in the revised supporting material, the TAG-measured SOA tracers produced in early generations (e.g., DHOPA, phthalic acid, pinic acid) correlated well with LO-OOA derived from AMS while those associated with later generation products (e.g., $C_{3-5}$ DCAs, $C_{3-5}$ hDCAs) had stronger correlations with MO-OOA. After further investigating OA compositions in $PM_1$ during different episodes, we also find that local episodes were characterized by higher mass proportions of POA and less aged SOA (LO-OOA) while the mix-influenced and transport episodes were associated with higher mass proportions of more aged SOA (MO-OOA), which is consistent with the observations from TAG. To further clarify how the AMS and TAG DATA were used and interconnected for differentiating episodic events, we have revised Figure S7 in the supporting material to present and compare OA compositions in $PM_1$ and $PM_{2.5}$ during different episodes.

5) There are instances where I feel confused when the authors refer to SOA and POA to link to sources of OA in the different episodes reported in this study (for example the role of $O_3$).

*Response:* Our data indicate $O_3$ oxidation played a relatively limited role in SOA formation during local episodes. We'd like to make a few points related to this. First, the high $NO_x$ concentrations as well as the high mass ratios of $NO/NO_2$ during local episodes likely have kept $O_3$ low and thus suppressed $O_3$ oxidation pathway (Table 2). Consequently, we observed more significant increases in mass concentrations of SOA markers formed via OH oxidation pathway compared with those formed via $O_3$ oxidation pathway. For example, DHOPA, which is a typical SOA product of monoaromatics with OH radicals, showed drastic increase in the mass concentration by 777%, in reference to non-episodic periods. In comparison, $C_9$ acids, which are typical oxidation products of fatty acids with $O_3$, their mass concentrations increased by 326% during local episodes in reference to non-episodic periods. And the mass concentrations of αPinT and βCaryT, which are oxidation products of biogenic VOCs with $O_3$, increased by 393% and 276% during local episodes in reference to non-episodic periods, respectively. Such contrasts between SOA products from OH-initiated vs $O_3$-initiated oxidation pathways appear to suggest that SOA formed during local episodes were more influenced by pathways other than ozonolysis (e.g., OH oxidation). We have added these explanations in the revised manuscript (Lines 289-296) to further clarify SOA formation during different episodes.

6) Lines 16-20: Please clarify if secondary and primary are dominating the OA. It seems to me that (line 17): secondary sources were important and in the next few lines the authors report that primary also are important sources!

*Response:* The claim in Line 17 that local episodes were significantly influenced by SOA is deduced from the mass variations of major chemical components in $PM_{2.5}$. That is, SOA overall had higher percentage proportions in $PM_{2.5}$ during local episodes compared with mix-influenced and transport episodes, while the latter two were characterized by significant higher mass incrementations in secondary inorganic ions (e.g., nitrate) in $PM_{2.5}$. When we further investigated OA compositions with the measurement data obtained from the TAG system, we find that SOA enhancements during local episodes were associated with sources from vehicle and cooking emissions. In other words, abundant precursors from local vehicles and cooking emissions greatly contributed to the formation of local SOA. Therefore, it is not odd that we also observed mass incrementations in hopanes, alkanes, and fatty acids during local episodes, which are typical POA markers for vehicle and cooking emissions. In this case, the claims in Lines 16-20 do not contradict each other.

7) Is sampling done every 2 hours (see abstract) or every one hour as mentioned in the text (Table 1/line 63 etc.)? Please clarify.

*Response:* The time resolution of TAG system is 2-hour. We have corrected related information in Table 1.

8) Line 17: suggests replacing "elevation" with "increase" or clarifying the sentence!

*Response:* We have rephrased the sentence in the revised manuscript (Line 17).

9) Line 35. Please correct the references (e.g., "L. Chen et al., 2017" should be "Chen et al. 2017"). Please check this throughout the manuscript.

*Response:* We have corrected the references in the revised manuscript (Lines 34, 46, 47, 90, 100, 148, 175, 240, 373, 398, 446).

10) Line 54: Add reference(s) to Recent studies....end of the sentence.
*Response:* This sentence has been deleted as we have revised the introduction section to give more discussions about the findings and limitations of previous field studies conducted in Shanghai (Lines 49-75).

11) Table 1. Please clarify which parameters were measured in the gas or particle phase and for how long?
*Response:* All parameters listed in Table 1 were measured from 25th November 2019 to 23rd January 2020 as stated in Line 89. In the column "parameter" of Table 1, we have stated that organic molecular markers, inorganic water-soluble ions, OC, EC, and 15 trace elements were measured in $PM_{2.5}$, and organics were measured in $PM_1$. In other words, these parameters were measured in the particle phase. For the parameter "$C_2$ - $C_{12}$ VOCs", they were measured in the gas phase and we have clarified this in the revised manuscript (Table 1).

12) Line 94: Table S1...to the end of the sentence. It seems to me that IS were also measured. The quantification of the 98 cpds was done using IS!
*Response:* We apologize that our wording was unclear. Below is a detailed explanation of how we quantify the 98 compounds and the role of internal standards (IS). We added a series of deuterated ISs in each sample introduced to the TAG system to compensate the matrix effects and other injection-to-injection variations (Gosetti et al., 2010; Wang et al., 2020). In our study, calibration curves were first established before using the TAG system to measure ambient samples. To be specific, 5 μL of ISs was mixed with 0-5 loops (5 uL/loop) of external standards and co-injected into CTD cell for GC-MS quantification. This yielded a five-point calibration curve for each analyte. Calibration curves were established by fitting the normalized peak areas of external standards to their corresponding IS with respective concentrations. During the ambient measurements, we also introduced 1 loop (5 μL) of IS in each aerosol sample. Then we calculated peak area ratios of target organic compounds against their corresponding IS (listed in Table S1) for each ambient sample and used the above-mentioned calibration curves to quantify their masses in real aerosol samples. We have added above information in the revised supporting material (Text S1) and more detail descriptions of the TAG calibration and quantification method have also been given in several of our published papers (He et al., 2020; Wang et al., 2020).

**Referee #3**

**Responses to comments from Referee #3:**

1) I think the introduction part is not sufficient to raise the scientific question. Numerous field studies have been conducted in Shanghai over the past decades. The limitations of these previous studies should be clearly discussed, and the ways in which this study can address these limitations need to be stated.
*Response:* Thanks for your suggestion. We have given more discussions on the findings and limitations of previous field studies conducted in Shanghai in the revised manuscript (Lines 65-75).

2) TAG is the key instrument of this paper. However, I did not see much information about its operation in this paper. For example, how much was the TD temperature? Was the derivatization agents used in TAG? If so, what are the agents?
*Response:* Although the operation of the TAG instrument was described in detail in our earlier publications, we agree with the reviewer that more essential operation information needs to be included in the current manuscript to facilitate readers' comprehension of our work.

In brief, the CTD temperature program was set to be firstly held at 45 °C for 2 min, then increased to 330 °C in 6 min, and lastly held at 330 °C for 12 min. During this thermal desorption step, polar organic compound in $PM_{2.5}$ deposit on the CTD underwent in-situ derivatization under a helium stream saturated with N-methyl-N-(trimethylsilyl) trifluoroacetamide (MSTFA). MSTFA was the derivatization agent used in this study. We have added above information in the revised manuscript (Lines 102-108) and more detailed information related to our TAG system is provided in our previously published papers (He et al., 2020; Wang et al., 2020; Zhu et al., 2021).

3) The same comments also apply to other instruments. The method section is too brief.

*Response:* Other instruments are more commonly deployed in field studies and better known to the atmospheric measurement community. The working principles of these instruments mentioned have been described in our previously published papers. For example, the performance of AMS during this field campaign is reported in Huang et al. (2021). Qiao et al. (2014) and Zhou et al. (2016) have described the working principles of OC/EC analyzer and MARGA adopted in this study and their performances during this field campaign is available in Zhou et al. (2022). In addition, detailed information related to the VOC measurements by the GC-FID adopted in this study is given in Liu et al. (2019) and Wang et al. (2015). We have added these references in the revised manuscript (Lines 94-97).

4) Table S1 lists the range and average concentrations of the 98 quantified organic compounds. How did the authors quantify their concentration? Through standard injection? How were these compounds identified? Is the identification method very reliable? Or it is just a search through NIST MS database? Such information should be clearly presented.

*Response:* Below is a detailed explanation of how we identify and quantify the 98 compounds and this information will be added to the supplementary information document. In our study, calibration curves were first established before using the TAG system to measure ambient samples. To be specific, 5 μL of ISs was mixed with 0-5 loops (5 uL/loop) of external standards and co-injected into CTD cell for GC-MS identification and quantification. This yielded a five-point calibration curve for each analyte. Calibration curves were established by fitting the normalized peak areas of external standards to their corresponding IS with respective concentrations. During the ambient measurements, 1 loop (5 μL) of IS was also injected into each aerosol sample by the auto-injection system equipped in the TAG. The target organic compounds in aerosol samples were identified by the retention time and mass spectrum, which were obtained from their authentic standards. Then we calculated peak area ratios of target organic compounds against their corresponding IS (listed in Table S1) for each ambient sample and used the above-mentioned calibration curves to quantify their masses in real aerosol samples. We have added above information in the revised supporting material (Text S1) and more details of the TAG calibration and quantification method have been given in several of our previously published papers (He et al., 2020; Wang et al., 2020).

5) TAG can operate in one-hour resolution. Why it was operated with a time resolution of 2 hours (Line 80)? In Table 2, it stated that TAG's time resolution was 1 hour. Which number is correct?

*Response:* Yes, the TAG system can be operated in one-hour resolution as reported by other papers (Isaacman et al., 2014), if a short GC program is adopted. Figure R1 (see below) shows the temperature status of three components in the TAG system. Note that during part of sample analysis stage, the CTD is not ready for sample collection mode. Sampling can only start when the CTD temperature has dropped to 35°C. As such, the time allocated to sample collection would be less than 1 hour if an hourly time resolution is adopted, such as in previous studies by Goldstein and coworkers. In our work, we have adopted a longer GC program and a full hour for sample collection. The combined time for each cycle of sample collection and analysis is 2 hours. We have corrected the related information in Table 1.

[Figure]

Figure R1. The temperature program of CTD, focusing trap (FT) and to-GC-arm in one analysis cycle. The next sample's start and end points are also marked, which coincide with the start and the end of the GC analysis. Adapted from He et al. (2020).

6) Line 265: it is stated that "transported $PM_1$ contained higher proportions of more-oxidized organic aerosol (MO-OOA) while less oxidized organic aerosol (LO-OOA) accounted for more $PM_1$ mass during local episodes." Can this statement also be supported by the AMS f44 trace?
*Response:* Yes, this statement is also supported by the AMS f44 tracer. We have added the below figure in the revised supporting material (Figure S8).

[Figure]

Figure R2. F44 as a function of MO-OOA concentration bins.

7) This paper uses a very traditional way to separate POC and SOC. But this study has AMS data. Why not just do a PMF analysis for AMS data and separate a SOM factor? I think this way is more accurate.
*Response:* We agree with the reviewer that AMS PMF is capable of separating POC and SOC using characteristic ion fragments. The reason why we use the OC/EC ratio method to estimate SOM and POM is that OC and EC were measured in $PM_{2.5}$ while AMS provides $PM_1$ measurements. Using POC and SOC derived from $PM_1$ to approximate those in $PM_{2.5}$ would introduce an uncertainty that is not straightforward in quantifying. We note that the AMS-derived source factors and POM, SOM estimated by OC/EC ratio method were well correlated (Figure S3 in the revised supporting material), indicating that the OC/EC ratio method applied in this work overall gave reasonable estimations of POM and SOM.

8) It seems that AMS PMF analysis was done and the results were included in Figure 3. But what are the details of this analysis. How many factors were used? How about other key parameters used in PMF? I would not report the PMF results without showing the key PMF parameters.
*Response:* Our recently published paper (Huang et al., 2021) has described the AMS PMF analysis in detail. In brief, a total of seven source factors were resolved by the PMF model based on AMS data collected during this period (see below in Figure R3). We have added this reference as well as the below figure in the revised supporting material (Figure S1) so that interested readers can get more detail information about the AMS PMF analysis.

[Figure]

Figure R3. (a) Time series, (b) mass spectral profiles, (c) diurnal variations and fractional contributions of the OA factors from the 7-factor solution of AMS-PMF analysis. Adapted from Huang et al. (2021) figure S2.

**Other changes in the changes-tracked revised manuscript:**

1) In Line 123, "Figure S1" has been changed to "Figure S2" accordingly.
2) In Line 125, "Figure S2" has been changed to "Figure S3" accordingly.
3) In Line 155, "Figure S3" has been changed to "Figure S4" accordingly.
4) In Line 156, "Figure S4" has been changed to "Figure S5" accordingly.
5) In Line 160, "Text S1" has been changed to "Text S2" accordingly.
6) In Line 194, AMS data from January 1st to 23rd January 2020 has been added in Figure 1 (g).
7) In Line 242, "Figure S1" has been changed to "Figure S2" and "Figure S5" has been changed to "Figure S6" accordingly.
8) In Line 255, "Figure S6" has been changed to "Figure S7" accordingly.
9) In Line 326, "Figure S7" has been changed to "Figure S9" accordingly.
10) In Line 345, "Text S2" has been changed to "Text S3" accordingly.
11) In Line 348, "Figure S10" has been changed to "Figure S12" accordingly.
12) In Line 374, "Figure S11" has been changed to "Figure S13" accordingly.
13) In Line 387, "Figure S11" has been changed to "Figure S13" accordingly.

14) In Line 399, "Figure S11" has been changed to "Figure S13" accordingly.
15) In Line 407, "Figure S12a" has been changed to "Figure S14a" accordingly.
16) In Line 413, "Figure S12b" has been changed to "Figure S14b" accordingly.
17) In Lines 509-511, the reference is deleted as we have revised the introduction section according to the referee's suggestion.
18) In Lines 537-539, the reference is deleted as we have revised the introduction section according to the referee's suggestion.
19) In Lines 545-548, the reference is deleted as we have revised the introduction section according to the referee's suggestion.
20) In Lines 578-579, the reference is deleted as we have revised the introduction section according to the referee's suggestion.
21) In Lines 628-630, the reference is deleted as we have revised the introduction section according to the referee's suggestion.
22) In Lines 659-661, the reference is deleted as we have revised the introduction section according to the referee's suggestion.
23) In Lines 669-671, the reference is deleted as we have revised the introduction section according to the referee's suggestion.
24) In Lines 690-693, one reference has been added accordingly.

**References:**
Bianchi, F., Kurtén, T., Riva, M., Mohr, C., Rissanen, M. P., Roldin, P., Berndt, T., Crounse, J. D., Wennberg, P. O., Mentel, T. F., Wildt, J., Junninen, H., Jokinen, T., Kulmala, M., Worsnop, D. R., Thornton, J. A., Donahue, N., Kjaergaard, H. G. and Ehn, M.: Highly Oxygenated Organic Molecules (HOM) from Gas-Phase Autoxidation Involving Peroxy Radicals: A Key Contributor to Atmospheric Aerosol, Chem. Rev., 119, 3472-3509, 10.1021/acs.chemrev.8b00395, 2019.

Ehn, M., Thornton, J. A., Kleist, E., Sipilä, M., Junninen, H., Pullinen, I., Springer, M., Rubach, F., Tillmann, R., Lee, B., Lopez-Hilfiker, F., Andres, S., Acir, I., Rissanen, M., Jokinen, T., Schobesberger, S., Kangasluoma, J., Kontkanen, J., Nieminen, T., Kurtén, T., Nielsen, L. B., Jørgensen, S., Kjaergaard, H. G., Canagaratna, M., Maso, M. D., Berndt, T., Petäjä, T., Wahner, A., Kerminen, V., Kulmala, M., Worsnop, D. R., Wildt, J. and Mentel, T. F.: A large source of low-volatility secondary organic aerosol, Nature, 506, 476-479, 10.1038/nature13032, 2014.

Ervens, B., Feingold, G., Frost, G. J. and Kreidenweis, S. M.: A modeling study of aqueous production of dicarboxylic acids: 1. Chemical pathways and speciated organic mass production, Journal of Geophysical Research; J.Geophys.Res, 109, D15205-n/a, 10.1029/2003JD004387, 2004.

Gao, Y., Wang, H., Zhang, X., Jing, S., Peng, Y., Qiao, L., Zhou, M., Huang, D. D., Wang, Q., Li, X., Li, L., Feng, J., Ma, Y., & Li, Y.: Estimating Secondary Organic Aerosol Production from Toluene Photochemistry in a Megacity of China, Environ. Sci. Technol., 53, 8664–8671, 10.1021/acs.est.9b00651, 2019.

Gosetti, F., Mazzucco, E., Zampieri, D. and Gennaro, M. C.: Signal suppression/enhancement in high-performance liquid chromatography tandem mass spectrometry, Journal of Chromatography A, 1217, 3929-3937, 10.1016/j.chroma.2009.11.060, 2010.

Hao, C., Shepson, P. B., Drummond, J. W. and Muthuramu, K.: Gas Chromatographic Detector for Selective and Sensitive Detection of Atmospheric Organic Nitrates, Anal. Chem. (Wash. ), 66, 3737-3743, 10.1021/ac00093a032, 1994.

He, X., Wang, Q., Huang, X. H. H., Huang, D. D., Zhou, M., Qiao, L., Zhu, S., Ma, Y., Wang, H., Li, L., Huang, C., Xu, W., Worsnop, D. R., Goldstein, A. H. and Yu, J. Z.: Hourly measurements of organic molecular markers in urban Shanghai, China: Observation of enhanced formation of secondary organic aerosol during particulate matter episodic periods, Atmospheric environment (1994), 240, 117807, 10.1016/j.atmosenv.2020.117807, 2020.

Huang, D. D., Zhu, S., An, J., Wang, Q., Qiao, L., Zhou, M., He, X., Ma, Y., Sun, Y., Huang, C., Yu, J. Z. and Zhang, Q.: Comparative Assessment of Cooking Emission Contributions to Urban Organic Aerosol Using Online Molecular Tracers and Aerosol Mass Spectrometry Measurements, Environ. Sci. Technol., 55, 14526–14535, 10.1021/acs.est.1c03280, 2021.

Isaacman, G., Kreisberg, N. M., Yee, L. D., Worton, D. R., Chan, A. W. H., Moss, J. A., Hering, S. V., & Goldstein, A. H.: Online derivatization for hourly measurements of gas- and particle-phase semi-volatile oxygenated organic compounds by thermal desorption aerosol gas chromatography (SV-TAG), Atmospheric Measurement Techniques, 7(12), 4417–4429, 10.5194/amt-7-4417-2014, 2014.

Jaoui, M., Lewandowski, M., Offenberg, J. H., Colon, M., Docherty, K. S. and Kleindienst, T. E.: Characterization of aerosol nitroaromatic compounds: Validation of an experimental method, J. Mass Spectrom., 53, 680-692, 10.1002/jms.4199, 2018.

Keehan, N. I., Brownwood, B., Marsavin, A., Day, D. A. and Fry, J. L.: A thermal-dissociation-cavity ring-down spectrometer (TD-CRDS) for the detection of organic nitrates in gas and particle phases, Atmospheric measurement techniques, 13, 6255-6269, 10.5194/amt-13-6255-2020, 2020.

Kristensen, K., Cui, T., Zhang, H., Gold, A., Glasius, M. and Surratt, J. D.: Dimers in α-pinene secondary organic aerosol: effect of hydroxyl radical, ozone, relative humidity and aerosol acidity, Atmospheric chemistry and physics, 14, 4201-4218, 10.5194/acp-14-4201-2014, 2014.

Liu, Y., Wang, H., Jing, S., Gao, Y., Peng, Y., Lou, S., Cheng, T., Tao, S., Li, L., Li, Y., Huang, D., Wang, Q., & An, J.: Characteristics and sources of volatile organic compounds (VOCs) in Shanghai during summer: Implications of regional transport, Atmospheric Environment (1994), 215, 116902–, 10.1016/j.atmosenv.2019.116902, 2019.

Lu, Y., Ma, Y., Huang, D. D., Lou, S., Jing, S., Gao, Y., Wang, H., Zhang, Y., Chen, H., Chang, Y., Yan, N., Chen, J., George, C., Riva, M. and Huang, C.: Unambiguous identification of N-containing oxygenated organic molecules using a chemical-ionization Orbitrap in an eastern Chinese megacity, Atmospheric chemistry and physics, 23, 3233-3245, 10.5194/acp-23-3233-2023, 2023.

Nie, W., Yan, C., Huang, D. D., Wang, Z., Liu, Y., Qiao, X., Guo, Y., Tian, L., Zheng, P., Xu, Z., Li, Y., Xu, Z., Qi, X., Sun, P., Wang, J., Zheng, F., Li, X., Yin, R., Dallenbach, K. R., Bianchi, F., Petäjä, T., Zhang, Y., Wang, M., Schervish, M., Wang, S., Qiao, L., Wang, Q., Zhou, M., Wang, H., Yu, C., Yao, D., Guo, H., Ye, P., Lee, S., Li, Y. J., Liu, Y., Chi, X., Kerminen, V., Ehn, M., Donahue, N. M., Wang, T., Huang, C., Kulmala, M., Worsnop, D., Jiang, J. and Ding, A.: Secondary organic aerosol formed by condensing anthropogenic vapours over China's megacities, Nature geoscience, 15, 255-261, 10.1038/s41561-022-00922-5, 2022.

Qiao, L., Cai, J., Wang, H., Wang, W., Zhou, M., Lou, S., Chen, R., Dai, H., Chen, C., and Kan, H.: PM2.5 constituents and hospital emergency-room visits in Shanghai, China, Environ. Sci. Technol., 48, 10406–10414, https://doi.org/10.1021/es501305k, 2014.

Szmigielski, R., Surratt, J. D., Gómez-González, Y., Van der Veken, P., Kourtchev, I., Vermeylen, R., Blockhuys, F., Jaoui, M., Kleindienst, T. E., Lewandowski, M., Offenberg, J. H., Edney, E. O., Seinfeld, J. H., Maenhaut, W. and Claeys, M.: 3-methyl-1,2,3-butanetricarboxylic acid: An atmospheric tracer for terpene secondary organic aerosol, Geophys. Res. Lett., 34, L24811-n/a, 10.1029/2007GL031338, 2007.

Wang, H. L., Qiao, L. P., Lou, S. R., Zhou, M., Chen, J. M., Wang, Q., Tao, S. K., Chen, C. H., Huang, H. Y., Li, L., & Huang, C.: PM$_{2.5}$ pollution episode and its contributors from 2011 to 2013 in urban Shanghai, China, Atmospheric Environment (1994), 123, 298–305, 10.1016/j.atmosenv.2015.08.018, 2015.

Wang, Q., He, X., Zhou, M., Huang, D. D., Qiao, L., Zhu, S., Ma, Y., Wang, H., Li, L., Huang, C., Huang, X. H. H., Xu, W., Worsnop, D., Goldstein, A. H., Guo, H. and Yu, J. Z.: Hourly Measurements of Organic Molecular Markers in Urban Shanghai, China: Primary Organic Aerosol Source Identification and Observation of Cooking Aerosol Aging, ACS Earth Space Chem, 4, 1670-1685, 10.1021/acsearthspacechem.0c00205, 2020.

YAN, M. A., RUSSELL, A. T. and MARSTON, G.: Mechanisms for the formation of secondary organic aerosol components from the gas-phase ozonolysis of α-pinene, Phys Chem Chem Phys, 10, 4294-4312, 10.1039/b803283a, 2008.

Yang, L., Ray, M. B. and Yu, L. E.: Photooxidation of dicarboxylic acids—Part II: Kinetics, intermediates and field observations, Atmospheric environment (1994), 42, 868-880, 10.1016/j.atmosenv.2007.10.030, 2008.

Zhang, J., He, X., Gao, Y., Zhu, S., Jing, S., Wang, H., Yu, J. Z., & Ying, Q.: Estimation of Aromatic Secondary Organic Aerosol Using a Molecular Tracer-A Chemical Transport Model Assessment, Environ. Sci. Technol., 55, 12882 – 12892, 10.1021/acs.est.1c03670, 2021.

Zhou, M., Qiao, L., Zhu, S., Li, L., Lou, S., Wang, H., Wang, Q., Tao, S., Huang, C., and Chen, C.: Chemical characteristics of fine particles and their impact on visibility impairment in Shanghai based on a 1-year period observation, J. Environ. Sci. (China), 48, 151–160, 10.1016/j.jes.2016.01.022, 2016.

Zhou, M., Zheng, G., Wang, H., Qiao, L., Zhu, S., Huang, D., An, J., Lou, S., Tao, S., Wang, Q., Yan, R., Ma, Y., Chen, C., Cheng, Y., Su, H., & Huang, C.: Long-term trends and drivers of aerosol pH in eastern China, Atmospheric Chemistry and Physics, 22(20), 13833–13844, 10.5194/acp-22-13833-2022, 2022.

Zhu, S., Wang, Q., Qiao, L., Zhou, M., Wang, S., Lou, S., Huang, D., Wang, Q., Jing, S., Wang, H., Chen, C., Huang, C. and Yu, J. Z.: Tracer-based characterization of source variations of PM$_{2.5}$ and organic carbon in Shanghai influenced by the COVID-19 lockdown, Faraday Discuss., 226, 112-137, 10.1039/D0FD00091D, 2021.

---

## Author Response (AR2)

**Manuscript title: Evolution and chemical characteristics of organic aerosols during wintertime PM$_{2.5}$ episodes in Shanghai, China: Insights gained from online measurements of organic molecular markers**

**Point-by-point response to comments**

**We appreciate all the comments and suggestions from the referees and editor. They are valuable in helping us improve our manuscript. Our point-by-point response is provided below and marked in blue text. Note the line numbers quoted in this response document correspond to the changes-tracked revised manuscript, which also show the revised text in blue.**

**Responses to comments from Referee #1:**

1) The authors responded properly to the comments raised by the reviewers and corresponding modifications were made to the manuscript. Though the revised manuscript was more clearly presented than the original version, I still have one more comment on the statement of more SOA contribution during the local episodes. It can be found from Table 2 that the average SOM/POM ratio was 4.0, 5.1, 3.9 and 3.8 during the local episodes, mixed-influence episodes, transport episodes and non-episodic periods respectively, it would thus be misleading to say " Episodes primarily influenced by local air masses were characterized with higher proportions and mass increments of secondary OA". Comparing with POM, SOM contribution during the local episodes was not significantly higher, but the contribution of inorganic ions significantly lower. I would suggest the authors to verify the concern and make corresponding modification to the manuscript.

   *Response:* The statement "more SOA contribution during the local episodes" is meant to describe "more SOA contributions to PM2.5", which is 26.5% during the local episode, higher than the other types of episodes (See Table 2). We'd like to clarify that it is meant to indicate that more SOM relative to POM (i.e., SOM/POM ratio) as understood by the reviewer. We now revise the statement to the following to improve the clarity:

   **Line 17:**

   "Episodes primarily influenced by local air masses were characterized with higher proportions in PM$_{2.5}$ and mass increments of both primary and secondary OA".

   In the main content when we discuss the variations of major components in PM$_{2.5}$ during different episodic events (Line 216-220), we have also mentioned that "Indeed, primary species (e.g., POM, EC, potassium, chloride, geological material matters and other trace elements) also showed noticeable increases with summed contributions up to 29% during local episodes, while their percent contributions during the transport and mixed-influence episodes were in the range of 8-14%. The higher proportions of primary species together with significantly higher values of NO/NO$_2$ and T/B ratios indicate that local PM$_{2.5}$ episodes in Shanghai were largely influenced by freshly emitted primary pollutants in the local areas".

**Responses to comments from Referee #3:**

1) During the last round of review, I asked the authors to add some key experimental details to the manuscript. It is important to let readers know the full details of the experiments without spending much effort to check the authors' previous papers. But only a few details were added in the revised paper. I think it is not sufficient.

   *Response:* In the revised manuscript (Line 79-93), we have added more information related to the operational procedure of the TAG system. Additionally, we have also given more descriptions for other instruments employed in this study in the revised supporting material (Text S1, Line 32-71).

   Line 79-94

   "The measurement principle and operational procedure of the TAG system have been detailed in previous studies (He et al., 2020; Kreisberg et al., 2009; Wang et al., 2020; Williams et al., 2006; S. Zhu et al., 2021). In brief, the TAG system was operated with a time resolution of 2 hours. During the first hour, aerosol was collected at a flow of 10 L/min, and during the second hour, GC-MS analysis was performed. After sampling at room temperature and subsequent addition of 5 µL internal standard (IS) mixtures, the thermal desorption cell (CTD) was held at 45 °C for

2 min, then increased to 330 °C in 6 min, and held at 330 °C for 12 min. During this thermal desorption step, polar organic compound in $PM_{2.5}$ deposit on the CTD underwent in-situ derivatization under a helium stream saturated with derivatization agent N-methyl-N-(trimethylsilyl) trifluoroacetamide (MSTFA). Subsequently, the organic compounds were re-concentrated onto a focusing trap (FT) cooled by a fan. Afterwards, the CTD was purged with pure helium to vent the excess MSTFA and the FT was heated to 330°C to transfer the organic compounds into the valve-less injection (VLI) system, which employs a restrictive capillary tube to connect with the inlet of the gas chromatograph (GC). Then the GC/MS analysis started and concurrently, the next ambient sample was collected via the above-mentioned steps. In this study, a total of 98 polar and nonpolar organic compounds were identified and quantified (Text S1) and the full list is provided in Table S1. The detailed quality control measures and results for the TAG measurements have also been reported in S. Zhu et al. (2021) and given in section 2.2.1."

Text S1, line 32-71

"An Aerosol Mass Spectrometer (AMS) was deployed to quantify major components in $PM_1$ during the campaign. The AMS was operated alternately between V & pToF combined mode and W mode for 150 s each. Filtered ambient air was sampled and analyzed before and after the campaign for 30 min with a HEPA-filter placed in front of the inlet, defined as the filter periods. The gas-phase $CO_2$ contribution to the $CO_2^+$ signal was corrected by the data during the filter periods and the detection limits of species are defined as three times the standard deviation of the measured species concentrations in the filter periods, which were 0.19, 0.033, 0.067, 0.182, and 0.032 μg/m$^3$ for organic, sulfate, nitrate, ammonium, and chloride, respectively. The PMF2 algorithm with a toolkit (version 3.04A) based on Igor Pro software was applied to perform PMF analysis of AMS mass spectra. More detail descriptions of AMS-PMF analysis during this measurement period have been reported by Huang et al (2021) and the PMF analysis results are given in Figure S1.

For volatile organic compounds, two on-line gas chromatograph with flame ionization detector (GC-FID) systems (Chromato-sud airmoVOC C2-C6 #5250308 and airmoVOC C6-C12 #2260308, Chromatotec, Bordeaux, France) were employed to provide their mass concentrations continuously with 30 min time resolution. The $C_2$ - $C_6$ VOCs were collected through a preconcentration trap containing porous substances (Carbotrap C, Carbopack B and Carboxen). The trap was cooled by a cell with Peltier effect during the sampling period. After sampling, it was heated to 220 ℃ to thermally desorb trapped $C_2$ - $C_6$ VOCs. For the $C_6$ - $C_{12}$ VOCs, they were collected with a trap filled with Carbotrap B, which was also cooled during the sampling period while heated to 380 ℃ during the thermal desorption step. The desorbed $C_2$ - $C_6$ and $C_6$ – $C_{12}$ VOC compounds were then separated on ultimetal column and quantified by flame ionization detector (FID). Calibrations were conducted automatically once a day with three internal permeation tubes containing standard compounds during the campaign. Additionally, manual calibrations by standard gas (Spectra, USA) were also performed before and after the campaign.

Major components and trace elements in $PM_{2.5}$ were measured in this study with hourly time-resolution. Among them, water-soluble inorganic ions, including $Cl^-$, $NO_3^-$, $SO_4^{2-}$, $Na^+$, $NH_4^+$, $K^+$, $Mg^{2+}$, and $Ca^{2+}$, were measured by a commercial instrument for online monitoring of aerosols and gases (MARGA, model ADI 2080, Applikon Analytical B.V.). In this instrument, aerosol samples were first drawn through a wet rotating annular denuder (WRD) where water-soluble gases diffused to the absorption solution (0.0035% $H_2O_2$), then particles were collected in a stream-jet aerosol collector (SJAC). After sampling, the absorption solutions were drawn from the WRD and the SJAC to syringes and subsequently injected to ion chromatographs with an internal standard (LiBr) for quantifications.

OC and EC in $PM_{2.5}$ were monitored using a semicontinuous OC/EC analyzer (model RT-4, Sunset Laboratory Inc.) equipped with a $PM_{2.5}$ cyclone and an upstream parallel-plate organic denuder (Sunset Laboratory Inc.). Ambient $PM_{2.5}$ was sampled on a quartz filter in the oven at a flow rate of 8.0 L/min. Then the sample was analyzed by the thermal-optical transmittance method (TOT) using a two-stage thermal procedure that consisted of 600 - 840 °C in a helium atmosphere and 550 - 650 - 870 °C in an oxidizing atmosphere (2% oxygen in helium).

A total of 15 trace elements (K, Ca, V, Cr, Mn, Fe, Ni, Cu, Zn, As, Se, Ba, Pb, Si, and S) in $PM_{2.5}$ were measured at the site using an online non-destructive X-ray fluorescence spectrometer (XRF, model Xact 625, Cooper Environmental), which employs a reel-to-reel method to sample and analyze elements. $PM_{2.5}$ samples were pumped through a section of Teflon filter tape at a flow rate of 16.7 L/min. Then the section of filter tape was analyzed by non-destructive X-Ray Fluorescence. The sampling and analysis processes occurred simultaneously, producing hourly data for the monitored trace elements."